# Speak-to-Structure: Evaluating LLMs in Open-domain Natural Language-Driven Molecule Generation

## Abstract

Recently, Large Language Models (LLMs) have shown great potential in natural language-driven molecule discovery. However, existing datasets and benchmarks for molecule-text alignment are predominantly built on a one-to-one mapping, measuring LLMs' ability to retrieve a single, pre-defined answer, rather than their creative potential to generate diverse, yet equally valid, molecular candidates. To address this critical gap, we propose Speak-to-Structure ($S^2$-**Bench**), the first benchmark to evaluate LLMs in open-domain natural language-driven molecule generation. $S^2$-Bench is specifically designed for one-to-many relationships, challenging LLMs to demonstrate genuine molecular understanding and generation capabilities. Our benchmark includes three key tasks: molecule editing (**MolEdit**), molecule optimization (**MolOpt**), and customized molecule generation (**MolCustom**), each probing a different aspect of molecule discovery. We also introduce **OpenMolIns**, a large-scale instruction tuning dataset that enables Llama-3.1-8B to surpass the most powerful LLMs like GPT-4o and Claude-3.5 on $S^2$-Bench. Our comprehensive evaluation of 30 LLMs shifts the focus from simple pattern recall to realistic molecular design, paving the way for more capable LLMs in natural language-driven molecule discovery.

## 1 Introduction

Molecule discovery plays a pivotal role in various scientific research fields, from pharmaceuticals (Keiser et al., 2010) to materials science (Higuchi et al., 2023). Traditionally, this is a trial-and-error process (Ekins, 2024) that requires extensive experimentation and data analysis (Mattern & Grosser, 2023), often taking over a decade to bring a new drug candidate to market (Lee et al., 2018).

Recently, Large Language Models (LLMs) have demonstrated great potential in molecule discovery (Edwards et al., 2021; 2022) by leveraging their powerful language understanding and robust reasoning abilities (OpenAI, 2023). Molecules can be represented as SMILES strings (Weininger, 1988), which allows LLMs to process molecules as textual strings, effectively bridging the gap between molecular structures and natural language. Meanwhile, by aligning molecules with textual descriptions (Edwards et al., 2022), LLMs can assist chemists in interpreting chemical knowledge, suggesting structural modifications, and predicting compound properties (Zhou et al., 2025a; Li et al., 2024a; Zhang et al., 2024), thereby significantly streamlining the molecule discovery process.

While the integration of LLMs holds immense promise, a significant challenge lies in the limitations of current datasets and benchmarks. For example, existing datasets for aligning molecules and texts like ChEBI-20 (Edwards et al., 2022) and PubChem324K (Liu et al., 2023), are constructed based on a one-to-one mapping assumption, where each textual description is linked to a single, predefined target molecule due to the convenience of acquiring ground truth data from existing databases. However, the one-to-one mapping presents a key limitation: it does not fully align with the nature of real-world molecule discovery. In practice, multiple distinct molecules can often share the same desired properties or biological activity (Petrone et al., 2012). For instance, a particular pharmaceutical effect is rarely unique to a single compound, and a material with a specific physical property, like tensile strength, can be realized through various molecular structures. This mismatch

Figure 1: Task illustration of $S^2$-Bench for open domain natural language-driven molecule generation. In contrast to text-based target molecule generation, multiple valid molecules may fulfill the textual requirements (right of the arrow).

between the evaluation paradigm and the reality of molecule discovery casts doubt on whether these datasets and benchmarks genuinely assess an LLM's capacity for creative molecular design, or if they instead inadvertently encourage models to rely on memorization and pattern-matching.

To bridge this critical gap, we introduce a novel benchmark, **S**peak-to-**S**tructure (**$S^2$-Bench**), the first benchmark designed to evaluate the open-domain natural language-driven molecule generation capabilities of LLMs. $S^2$-Bench is meticulously structured into three primary tasks, as shown in Figure 1, each targeting a specific, real-world capability essential for drug and materials science:

**Molecule Editing (MolEdit)** focuses on testing an LLM's ability to perform precise, localized structural modifications while preserving main structures. In drug discovery, this is analogous to Lead Optimization (Barcelos et al., 2022), which edits the lead compound to create new variants. This requires a deep understanding of chemical grammar, including valency, stereochemistry, and ring systems, ensuring that edits could create valid molecules.

**Molecule Optimization (MolOpt)** extends the MolEdit task by evaluating a model's ability to optimize a lead compound under specified property constraints (e.g., increasing solubility or reducing toxicity). Its open-ended design requires the model to edit and refine molecular structures in accordance with the desired properties, thereby showcasing its understanding of both structural modifications and property-related characteristics.

**Customized Molecule Generation (MolCustom)** requires LLMs to synthesize a novel molecule based on a natural language description of its desired structural components. Specifically, it challenges models to generate molecules with a set of quantitative and qualitative constraints, such as a specific number of atoms, bonds, or functional groups, which directly mimics the initial drug design phase where a chemist defines a new compound based on a set of precise requirements for its structure.

To facilitate the evaluation of the above tasks, we also introduce a robust evaluation system with tailored examination process, enabling a comprehensive assessment of LLMs' performance in open-domain natural language-driven molecule discovery. Furthermore, we propose **OpenMolIns**, a large-scale instruction-tuning dataset, comprising up to 120 million instruction–molecule pairs, enabling Llama3.1-8B to achieve the state-of-the-art performance among 30 powerful LLMs on $S^2$-Bench.

To summarize, our contributions are primarily threefold:

1. We introduce **S**peak-to-**S**tructure (**$S^2$-Bench**), the first benchmark for open-domain natural language-driven molecule generation. Moving from one-to-one to one-to-many relationships, $S^2$-Bench better aligns concepts in drug discovery and provides a novel perspective on assessing LLMs' genuine molecular understanding and design capabilities.

2. We introduce **OpenMolIns**, an instruction-tuning dataset with up to 120 million instruction–molecule pairs, enabling Llama3.1-8B to achieve SOTA performance on $S^2$-Bench.

3. We provide insightful findings based on our extensive benchmarking of 30 LLMs, revealing the limitations of existing targeted generation datasets and highlighting the potential for LLMs to transition from simple pattern recall to realistic molecular design.

## 2 RELATED WORK

Molecule discovery is a cornerstone of scientific progress, underpinning advances in both drug development and material design (Du et al., 2022). The integration of artificial intelligence into this process has driven a paradigm shift in the pharmaceutical landscape, substantially improving the efficiency and accuracy of identifying and developing novel therapeutic candidates.

Recent breakthroughs in natural language processing (NLP) have highlighted the potential of Large Language Models (LLMs) to analyze complex biological and chemical data more effectively than traditional computational methods (Zhou et al., 2023). Within this context, Text-based Molecule Generation has emerged as a representative task for AI-driven molecule discovery (Edwards et al., 2021). This task focuses on generating target molecules directly from natural language descriptions, which requires the construction of paired datasets linking molecular structures with their textual representations.

Early work in this direction employed transformer-based models such as MolT5 (Edwards et al., 2022), which leveraged large-scale self-supervised learning to produce high-quality SMILES strings from textual prompts. Building on this foundation, models like KV-PLM (Zeng et al., 2022), MoMu (Su et al., 2022), and BioT5 (Pei et al., 2023) integrated molecular graphs and biochemical texts to jointly enhance molecular understanding and generation.

Parallel to these architecture-driven improvements, LLMs such as MolReGPT (Li et al., 2024b) and ICMA (Li et al., 2024a) have demonstrated strong in-context learning capabilities, enabling adaptive molecule generation by retrieving and utilizing relevant examples from the provided context. Besides, MolReFlect (Li et al., 2024c), emphasizes fine-grained alignment between molecular structures and natural language descriptions through a teacher–student paradigm, which effectively captures subtle structural–semantic relationships. Moreover, Mol-R1 (Li et al., 2025) takes a step further by introducing DeepSeek-R1 like Long-CoT reasoning in molecule generation, enabling robust and interpretable natural language-driven molecule generation.

Meanwhile, Nicolaou & Brown (2013) propose the task of multi-objective molecular optimization and Wu et al. (2024) further extend it to textual requirements by using prompt engineering to guide LLM-based molecule optimization. In addition, Ye et al. (2025) incorporate explicit structural constraints into the optimization process, while InstructMol (Zhuang et al., 2025) further proposes an LLM that can understand and design biomolecules following human instructions. Taking a step further, Zhou et al. (2025b) propose a teacher–student framework and the multi-constrained molecular generation problem, in which a teacher model distills molecular knowledge (structure, properties, and binding affinity) into natural language, and a student model is trained to satisfy multiple predefined constraints simultaneously. This paradigm focuses primarily on property-driven optimization under a fixed constraint set. Our work differs from this previous work in several important aspects:

1. **Motivation.** $S^2$-Bench target the **open-ended** nature of textual molecular generation. Rather than optimizing for a fixed list of constraints, our benchmark evaluates whether an LLM can generate any molecule that satisfies arbitrary user-specified textual requirements, covering customized generation (MolCustom), structural editability (MolEdit), and optimization tasks (MolOpt).

2. **Task Scope.** Our tasks require models to understand molecular structure at a fine-grained level, including atoms, bonds, stereochemistry, and functional groups, and to perform precise modifications that directly affect molecular properties. In contrast, the concurrent work centers on functional-group descriptions and property estimation.

3. **Dataset Construction.** Our dataset is fully template- and toolkit-driven, with instructions generated from a diverse template pool and evaluation performed automatically using cheminformatics tools such as RDKit. The concurrent work relies on a teacher model to generate natural-language descriptions from molecular data, forming a knowledge-distillation pipeline rather than a template-driven benchmark.

4. **Contributions.** $S^2$-Bench provides a novel benchmark, an automatic evaluation protocol, and systematic insights into LLMs' molecular understanding. The concurrent work contributes a new model architecture and a large-scale text–molecule dataset tailored for multi-constrained optimization.

# 3 SPEAK-TO-STRUCTURE ($S^2$-BENCH)

In this section, we detail the design philosophy, technical composition, and statistics of Speak-to-Structure (**$S^2$-Bench**), which is fundamentally structured around the core capabilities required for real-world molecular design: editing, optimizing, and customized generation.

## 3.1 TASK COMPOSITION

$S^2$-Bench is meticulously designed to move beyond simple pattern-matching and evaluate the genuine molecular understanding and generation capabilities of LLMs. We adopt a one-to-many paradigm, where a single language instruction can be fulfilled by multiple valid molecules. Our benchmark challenges LLMs to demonstrate a flexible understanding of molecular syntax rather than mere memorization. This design philosophy is concretely realized through three tasks of increasing complexity, each mirroring a critical phase of molecule discovery.

**MolEdit** assesses an LLM's foundational molecular syntax and structural modification capabilities. In this task, a base molecule is provided, and the model is asked to perform a specific change on the molecular structure. In a real-world scenario, this task is analogous to the iterative lead optimization process in drug discovery, where chemists make small, precise changes to a candidate molecule to change its properties. We designed three subtasks to probe this capability: *AddComponent* and *DelComponent* test the model's ability to add or remove a specific functional group, respectively, while *SubComponent* challenges it to perform a combined operation of both.

**MolOpt** evaluates an LLM's ability to perform goal-oriented chemical reasoning. The challenge here is twofold: the LLM should not only edit a molecule but also ensure that the modification leads to a desired change in a specific property. This task directly mirrors the hit-to-lead optimization stage, where molecules are refined to achieve better pharmacological profiles. To assess this, we chose three key properties (*LogP*, *MR*, and *QED*), which are vital for drug discovery and can be **reliably and reproducibly calculated**. By requiring the model to generate a molecule that improves on a given property, we test its ability to understand the complex relationship between molecular structure and function, a capability that cannot be evaluated by tasks with a single, predetermined answer.

**MolCustom** serves as the ultimate test of an LLM's creative chemical design ability. MolCustom requires the LLM to generate a novel molecule from a natural language description of its desired structural components. This task is not about modifying an existing structure but about synthesizing a new one from scratch, akin to a chemist defining the needs for a new compound at the very beginning of a project. To ensure a clear and precise evaluation of this complex task, we focus on generating molecules with specific structural features: a defined number of atoms (*AtomNum*), a specified number and type of bonds (*BondNum*), or the inclusion of particular functional groups (*FunctionalGroup*). While these constraints may seem simple, they are deceptively challenging, requiring the model to apply a sophisticated understanding of chemical valence rules to create a valid and novel molecule.

Compared to targeted molecule generation datasets, these tasks in $S^2$-Bench, while seemingly straightforward in their instructions, impose a more rigorous requirement on LLMs' ability to perform precise, open-domain molecule generation. Our benchmark fundamentally shifts the evaluation from pattern recall to practical chemical design, thereby providing a more accurate measure of an LLM's potential in molecule discovery and helping to build more explainable and trustworthy models.

## 3.2 DATA CONSTRUCTION

Previously, the development of robust datasets for text-based targeted molecule generation has been hindered by the scarcity of high-quality human annotations. For example, while text-to-image generation datasets like MS COCO (Chen et al., 2015) contain millions of annotated samples, molecule-caption datasets, such as ChEBI-20 (Edwards et al., 2022) are significantly smaller, often by orders of magnitude. This disparity arises because molecular annotation demands specialized expertise, making it both time-intensive and costly. The resulting data scarcity further poses a significant challenge for advancing LLMs in text-guided molecule discovery.

Our approach to data construction fundamentally bypasses this bottleneck. $S^2$-Bench is designed for open-domain molecule generation tasks that do not rely on human annotations. Instead, we leverage automated chemical toolkits to programmatically construct tasks and evaluate outputs based on

objective molecular properties and structural rules. This enables us to generate a virtually unlimited volume of data, effectively addressing the "data hunger" that has constrained previous research. More importantly, this programmatic approach allows us to create tasks with a one-to-many relationship, where multiple correct answers exist for a single prompt, which better aligns with the complexities of drug discovery, accommodating its inherent variability and diverse outcomes.

Specifically, we design a systematic and scalable data construction process, as illustrated in Figure 4.

For **MolEdit** and **MolOpt**, we sample molecules from large, publicly available chemical databases. We selected Zinc-250K (Sterling & Irwin, 2015) for building our test set due to its manageable size and diversity, while the massive PubChem (Kim et al., 2019) database (with over 10 million molecules) serves as the basis for the instruction-tuning dataset. We utilize the RDKit (Landrum, 2013) toolbox to automatically extract key molecular statistics, including structural patterns and properties like LogP, MR, and QED. These extracted features are then integrated into our pre-defined, instruction-based prompt templates, which is demonstrated in Appendix C.

For **MolCustom**, we move beyond existing databases and programmatically generate instructions to test a model's ability to create novel molecules. For each of the three subtasks, we randomly generate 5,000 instructions that specify a target number and type of atoms, bonds, or functional groups.

Furthermore, for each subtask, we pre-defined a diverse prompt template pool to ensure that the LLMs are not overfit to a limited set of prompt formats. This programmatic construction not only allows for the generation of a much larger data volume but also ensures that our benchmark tests an LLM's intrinsic chemical reasoning ability, rather than its capacity for memorizing human-annotated patterns. By moving beyond the one-to-one paradigm, $S^2$-Bench provides a more robust and realistic evaluation, paving the way for models that can truly innovate and design novel molecules.

## 3.3 EVALUATION

The evaluation of $S^2$-Bench is facilitated through a set of carefully designed automated evaluation processes and metrics tailored to the unique nature of our tasks. Unlike one-to-one datasets or benchmarks that simply check for exact matches, our metrics are designed to assess an LLM's ability to produce valid, relevant, and novel outputs within an open-ended framework.

| Task | (Weighted) Success Rate | Similarity | Novelty | Validity |
|------|:-:|:-:|:-:|:-:|
| MolEdit | ✓ | ✓ | | ✓ |
| MolCustom | ✓ | ✓ | | ✓ |
| MolOpt | ✓ | | ✓ | ✓ |

Table 1: All used metrics are listed here. '✓' means that a metric is calculated on a task.

### 3.3.1 MOLEDIT & MOLOPT EVALUATION

For the **MolEdit** and **MolOpt** tasks, as shown in Table 1, we employ a combination of metrics to assess both the correctness of the modification and the rationality of the design.

**Success Rate**: This metric evaluates a model's ability to fulfill the specific molecule generation requirements, with values ranging from 0 to 1. We design automated evaluation processes to verify if the generated molecule meets the specified criteria (e.g., correct functional group modification for MolEdit, or desired property optimization for MolOpt). Detailed Implementations of the automated testing process are shown in Appendix D. This metric examines whether the LLM can follow instructions precisely, which is a fundamental requirement for an LLM to serve as a chemist assistant.

**Similarity**: In open-domain tasks like MolEdit and MolOpt, a high Success Rate alone is insufficient. We must also ensure that the generated molecule is a reasonable modification of the original, rather than a completely new, unrelated structure that happens to satisfy the criteria. To address this, we measure the Tanimoto Similarity between the generated and original molecules using Morgan Fingerprints (Butina, 1999). A high similarity score indicates that the model has performed a rational, localized edit and has not simply generated a different molecule from scratch. The similarity $\delta(m^g, m^o) \in [0, 1]$ between the generated molecule $m^g$ and the original molecule $m^o$ is calculated as:

$$\delta(m^g, m^o) = \frac{|fp_{m^g} \cap fp_{m^o}|}{|fp_{m^g} \cup fp_{m^o}|}, \tag{1}$$

where $fp_{m^g}$ and $fp_{m^o}$ represents their corresponding Morgan Fingerprints. $|fp_{m^g} \cap fp_{m^o}|$ is the size of the intersection between their Morgan Fingerprints, while $|fp_{m^g} \cup fp_{m^o}|$ is the union.

**Validity**: This metric evaluates the percentage that the generated molecules are chemically valid and follow the rules of molecular syntax. Molecules need to successfully pass the SMILES parser, and a higher validity means that the model is more familiar with the molecule syntax.

### 3.3.2 MolCustom Evaluation

For the **MolCustom** task, where models are required to generate molecules from scratch based on structural descriptions, we use a different set of metrics to evaluate their creative design capabilities.

**Success Rate**: This metric measures how well the generated molecules adhere to the high-level structural constraints provided in the prompt (e.g., number of atoms, bonds, or specific functional groups). As the task requires de novo generation, the success rate directly reflects a model's ability to translate abstract requirements into a valid, concrete molecular structure. Detailed testing process of MolCustom could also be found in Appendix D.

**Novelty**: For open-domain generation, fulfilling the requirements is just the first step. The true value lies in a model's ability to generate novel and innovative molecules. The novelty score quantifies this by comparing the generated molecules against existing structures in a large database, such as Zinc-250K (Zinc for short). A low similarity to known molecules suggests a high degree of novelty, which is a critical indicator of a model's potential for discovering truly new molecule structures. The novelty $n$ for the generated molecule $m^g$ can be calculated as:

$$n(m^g) = 1 - \frac{\sum_{m^k \in Zinc} \delta(m^g, m^k)}{|Zinc|},$$ (2)

**Validity**: As with the other two tasks, validity ensures that all generated molecules are chemically sound and can be interpreted correctly.

### 3.3.3 Average Weighted Success Rate

To provide a single, comprehensive ranking of LLM performance on $S^2$-Bench, we introduce a weighted success rate. This metric combines the core success rate with a quality metric relevant to each task: Similarity for MolEdit/MolOpt and Novelty for MolCustom. This approach ensures that a high score reflects not only the ability to follow instructions but also the rationality and creativity of the generation. The weighted success rate for a subtask $t$ is defined as:

$$WSR_t = \begin{cases} n_t \times SR_t, & t \in \{MolCustom\} \\ \delta_t \times SR_t, & t \in \{MolEdit, MolOpt\} \end{cases},$$ (3)

where $WSR_t$ denotes the weighted success rate for a subtask $t$, while $\delta_t$ is the similarity score for the MolEdit and MolOpt tasks, $n_t$ represents the novelty score for the MolCustom tasks, and $SR_t$ is the corresponding success rate of the subtask. Then, the average weighted success rate $\overline{WSR}$ could evaluate the synthetic performance of LLMs among all the nine subtasks:

$$\overline{WSR} = \frac{1}{9} \sum_t WSR_t,$$ (4)

where the average weighted success rate $\overline{WSR}$ provides a balanced measure of a model's performance across all nine subtasks, offering a robust and nuanced assessment of its capabilities in open-domain natural language-driven molecule generation.

### 3.4 OpenMolIns: Instruction Tuning Dataset

To effectively train and evaluate LLMs on the open-ended challenges posed by $S^2$-Bench, we introduce **OpenMolIns**, a specialized instruction-tuning dataset derived from the PubChem database. The core philosophy behind OpenMolIns is to provide models with the kind of flexible, one-to-many training examples they need to learn genuine chemical reasoning, rather than simply memorizing specific input-output pairs. To ensure the integrity of our evaluations, we meticulously designed OpenMolIns

to have zero overlap with the Zinc-250K, preventing any data leakage that could compromise the validity of our benchmark results.

We built the instruction-tuning dataset programmatically, creating equal numbers of samples (i.e., 5,000) for all nine subtasks. The RDKit toolbox was essential to this process, allowing us to automatically generate data based on objective molecular properties and rules. This contrasts sharply with previous methods that rely on scarce and costly human annotations. Details of the construction process and the pre-defined prompt templates are shown in Appendix C.

For **MolEdit** and **MolOpt**, our goal is to improve the capabilities of LLMs to perform rational, goal-oriented modifications. For MolEdit, we start with a molecule, programmatically identify a modifiable functional group, and then perform a change (add, delete, or substitute) to create a new, valid molecule. This pair of (original and modified) molecules is then used to construct an instruction. This process teaches the model not only a single correct answer, but rather the skill of modifying a molecule in a chemically sound way. Similarly, for MolOpt, we calculate properties before and after a modification to identify a specific optimization direction (e.g., increasing LogP or decreasing MR). The training sample then explicitly links a desired outcome to a structural change, teaching the LLM to reason about the relationship between structure and property.

In the **MolCustom** domain, we aim to train the model to generate molecules from scratch based on structural details. Instead of relying on existing molecule-caption pairs, we programmatically analyze molecules from PubChem, extract key structural statistics (e.g., atom counts, bond numbers, and functional group types), and use these as the basis for a training prompt. The molecule itself becomes the target output. This approach trains the model to synthesize a new molecule that fits a set of high-level, multi-faceted criteria, a skill fundamentally different from recalling a specific, pre-existing structure. This method allows the generated molecules to better fit the real distribution of molecular space and prepares the model for de novo design.

To further investigate the impact of data scales on an LLM's ability to learn these tasks, we created five distinct data levels: light, small, medium, large, and xlarge, shown in Table 2. This tiered structure allows researchers to systematically analyze the data scaling law for open-domain natural language-driven molecule generation, providing valuable insights into how data quantity influences a model's capacity for chemical reasoning.

| Dataset | S$^2$-Bench | | OpenMolIns | | | | |
|---|---|---|---|---|---|---|---|
| item | each subtask | total | light | small | medium | large | xlarge |
| Data Size | 5,000 | 45,000 | 4,500 | 18,000 | 45,000 | 90,000 | 1,200,000 |

Table 2: Statisics of S$^2$-Bench and OpenMolIns.

## 3.5 STATISTICS

In this section, we provide a detailed overview of S$^2$-Bench and OpenMolIns, as summarized in Tables 2 and 3. The design and scale of our datasets are not arbitrary; they are meticulously crafted to enable a rigorous and nuanced evaluation of LLMs in open-domain molecule generation, which previous, smaller datasets cannot provide.

S$^2$-Bench is composed of nine subtasks, with each containing 5,000 test samples, for a total of 45,000 carefully curated test cases. This extensive size is critical for providing a statistically robust and comprehensive assessment of model performance, mitigating the risk of misleading results that can arise from smaller test sets. More importantly, these samples are designed to test the model's ability to handle one-to-many relationships, a challenge that is central to true chemical reasoning.

Meanwhile, to support the development of models capable of succeeding on this benchmark, OpenMolIns is provided in five distinct data scales, ranging from 4,500 to 1,200,000 examples. This tiered structure serves a dual purpose: it not only offers ample data to train powerful models but also allows researchers to systematically investigate the data scaling law for this novel task. By observing how model performance improves with increasing data size, we can gain valuable insights into the data requirements for aligning molecular space with natural language.

Besides, as shown in Table 3, S$^2$-Bench and OpenMolIns stand out from existing text-based molecule generation datasets. S$^2$-Bench is the first to introduce a truly open-domain natural language-driven molecule generation task, moving the field beyond the limitations of one-to-one mapping. In terms of

| Benchmarks or Datasets | Task Type | # Training Data | # Test Data | Public Access |
|---|---|---|---|---|
| PCdes(Zeng et al., 2022) | Targeted Generation | 10,500 | 3000 | ✗ |
| ChEBI-20 (Edwards et al., 2021) | Targeted Generation | 26,407 | 3,000 | ✓ |
| PubChem (Liu et al., 2023) | Targeted Generation | 12,000 | 2,000 | ✓ |
| Mol-Instructions (Fang et al., 2024) | Targeted Generation | 297,319 | 1000 | ✓ |
| L + M - 24 (Edwards et al., 2024) | Targeted Generation | 160,492 | 21,839 | ✓ |
| **S²-Bench & OpenMolIns** | Open Generation | 1,200,000 | 45,000 | ✓ |

Table 3: Comparison with existing text-based molecule generation benchmarks and datasets. Notably, we denote the statistics of the description guided molecule design part in Mol-Instructions Fang et al. (2024).

data scale, our datasets are also significantly larger. OpenMolIns has the largest training set to date, with a volume that is unprecedented in this domain. This scale is crucial for enabling LLMs to learn complex chemical rules and generalize beyond the memorization of specific examples. Similarly, S²-Bench's 45,000 test samples make it the largest evaluation benchmark, providing a more reliable and comprehensive measure of a model's true capabilities. This combination of innovative task design and large-scale data represents a new paradigm for evaluating and advancing LLMs in the field of text-guided molecular discovery.

# 4 EXPERIMENTS

## 4.1 MODELS

The models benchmarked are categorized into four groups: proprietary models, open-source general LLMs, open-source ChEBI-20 fine-tuned LLMs, and OpenMolIns fine-tuned LLMs.

**Proprietary Models.** This category includes LLMs that are only accessible via commercial API services. In this work, we benchmark GPT-4o, GPT-4-turbo, GPT-3.5-turbo (OpenAI, 2023), Claude-3.5 (Anthropic, 2024b), Claude-3 (Anthropic, 2024a), and Gemini-1.5-pro (Deepmind, 2024), which are all the most advanced LLMs with powerful reasoning and generalization capabilities. These LLMs are pre-trained on vast pre-training corpora, normally including chemical-related knowledge.

**Open-source General & Chemical LLMs.** This group contains open-source general LLMs with instruction following capability for a wide range of tasks, including Llama3-70B-Instruct, Llama3-8B-Instruct, Llama3.1-8B-Instruct, Llama3.2-1B-Instruct (Dubey et al., 2024), Mistral-7B-Instruct-v0.2 (Jiang et al., 2023), Deepseek-R1-distill-Qwen-7B (Guo et al., 2025), Qwen2-7B-Instruct (Yang et al., 2024), yi-1.5-9B (Young et al., 2024), chatglm-9B (GLM et al., 2024), Gemma3-12B (Team et al., 2025). Additionally, domain-specific LLMs like ChemLLM-20B (Zhang et al., 2024) and ChemDFM-v1.5-8B (Zhao et al., 2025) are also considered.

**Open-source ChEBI-20 Fine-tuned LLMs.** LLMs fine-tuned on the ChEBI-20 dataset can grasp some extent of text-based molecule generation capability. In this case, our experiments also cover LLMs like MolT5-small, MolT5-base, MolT5-large (Edwards et al., 2022), and BioT5-base (Pei et al., 2023). These models have been the state-of-art models in the molecule-caption translation task. Here, to ensure a fair evaluation of the intrinsic capabilities of LLMs, we exclude models that utilize multi-modal architectures or retrieval-augmented generation.

**OpenMolIns Fine-tuned LLMs.** We further fine-tune LLMs like Galactica-125M (Taylor et al., 2022), Llama3.2-1B-Instruct, and Llama3.1-8B-Instruct on OpenMolIns dataset for comparison. We specifically include the experiments on five distinct data sizes of OpenMolIns for Galactica-125M because the model has been pre-trained on scientific corpora and has proved its effectiveness in molecule-related tasks (Liu et al., 2023). Meanwhile, a small size model can also help us study the data scaling law within reasonable budget. Furthermore, Llama3.2-1B-Instruct and Llama3.1-8B-Instruct are also selected due to their advanced capabilities and similar architectures.

## 4.2 FINDINGS

Based on our comprehensive benchmarking, we have made the following key observations regarding the capabilities of LLMs in open-domain natural language-driven molecule generation.

**F1: Current LLMs show promise but lack the structural understanding necessary for precise molecule generation.** As shown in Figure 2, top-tier proprietary models such as Claude-3.5 and Gemini-1.5-Pro attain 35.92% and 34.80% in average weighted success rate, respectively. These results suggest that general LLMs hold promise for open-ended molecule generation due to their

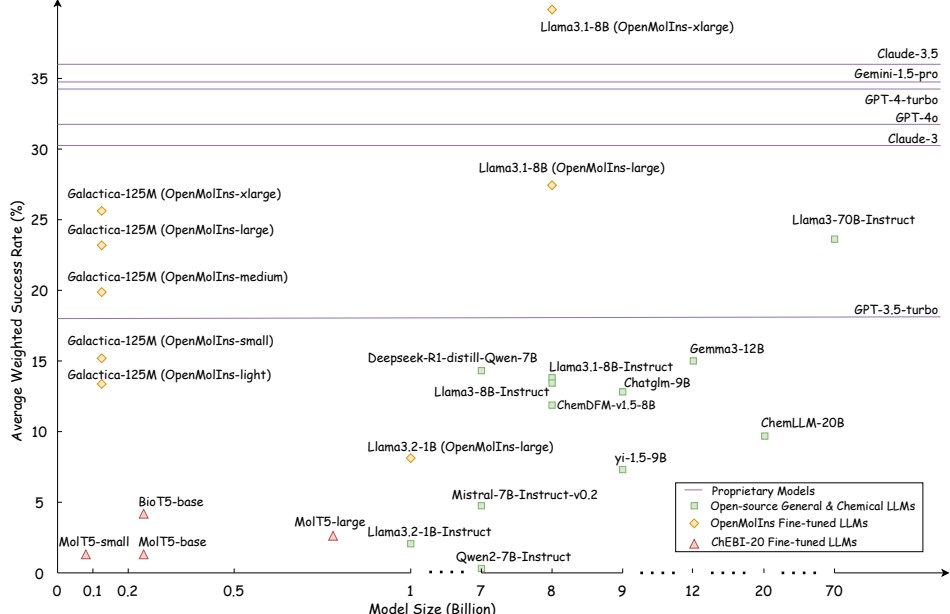

Figure 2: **The performance of LLMs benchmarked in S²-Bench.** LLMs fall into 4 categories: Proprietary Models, Open-source General LLMs, Open-source ChEBI-20 Fine-tuned LLMs, and OpenMolIns Fine-tuned LLMs. Models of unknown parameters are denoted as horizontal lines.

powerful reasoning capability, yet substantial challenges remain in translating textual requirements into chemically valid and structurally precise molecules. Notably, within the MolCustom task, no LLM manages to surpass a 25% weighted success rate on any individual subtask. The particularly poor performance on MolCustom tasks highlights that while current LLMs can generate chemically plausible molecules in a broad sense, they struggle with satisfying fine-grained structural constraints such as precise atom, bond, or functional group counts, because the precise control of molecular structures requires reasoning over discrete, globally coupled constraints. LLMs, even trained on SMILES, primarily learn local token distributions from large-scale pre-training data, which rarely emphasizes exact numeric constraints.

The precise control, in fact, is crucial for molecular discovery, as many properties of interest, such as solubility, toxicity, and active site composition, depend on specific structural configurations. Our finding suggests that open-domain natural language–driven molecule generation task can serve as an effective diagnostic tool, revealing systematic weaknesses in current LLMs and providing rich failure cases that can inform future instruction tuning or reinforcement learning strategies to improve molecule structural constraint adherence.

**F2: While pre-training provides a foundation for LLM performance, instruction tuning is indispensable for guiding and optimizing their capabilities in molecular discovery.** Open-source general LLMs can often lack chemical corpora during their pre-training stage. As a result, models like Qwen2-7B-Instruct and yi-1.5-9B obtained unsatisfactory results. The observed shortcoming can be attributed to the lack of chemical data in pre-training, despite their notable proficiency in mathematics and general language understanding. Surprisingly, open-source chemical LLMs such as ChemLLM-20B and ChemDFM-v1.5-8B achieve relatively poor performance, indicating that current chemical LLM pretraining still fails to capture the molecular structural details. In contrast, Llama3.1-8B-Instruct delivers superior performance at a relatively modest model scale, demonstrating a more balanced and comprehensive capability across diverse domains.

Meanwhile, our experiments with OpenMolIns highlight the effectiveness of instruction tuning. Fine-tuned on the largest OpenMolIns scale (i.e., xlarge), Llama3.1-8B-Instruct surpassed all other LLMs with an average weighted success rates of 39.33%. More remarkably, the much smaller model, Galactica-125M, within just 125 million parameters, achieved a weighted average success rate of 25.73%, outperforming models two orders of magnitude larger, such as Llama3-70B-Instruct. This remarkable outcome indicates that task-specific instruction tuning on a large, high-quality corpus can be highly effective, suggesting a practical avenue for LLMs to efficiently narrow the performance gap in molecular discovery.

**F3: One-to-one mappings between language instructions and molecules make LLMs inherently rely on pattern recognition and recall.** Our results also substantiate the observation, demonstrating

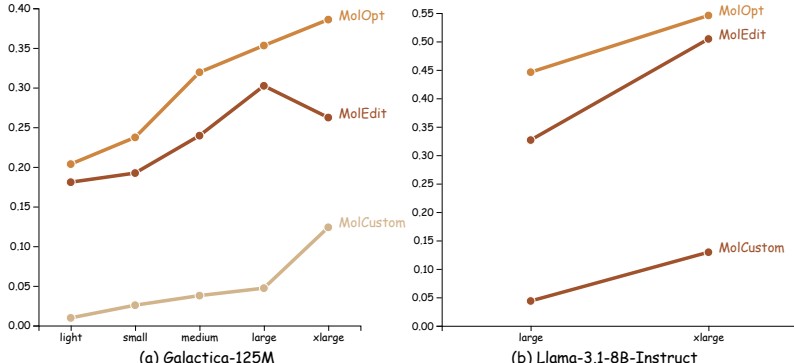

Figure 3: Task-specific performance scaling with increasing data in $S^2$-Bench. (a) Galactica-125M. (b) Llama-3.1-8B-Instruct.

that existing one-to-one mappings are insufficient for evaluating true chemical comprehension of LLMs. Notably, ChEBI-20 Fine-tuned LLMs, which are designed to align molecules with texts, perform worse than general LLMs. For instance, BioT5-base (Pei et al., 2023), a state-of-the-art model on the ChEBI-20 dataset, achieves an average weighted success rate of only 4.21% on $S^2$-Bench.

A closer examination of BioT5-base's performance provides insight into this limitation. Although the model achieves relatively high success rates on MolEdit and MolOpt tasks, its generated molecules often exhibit low similarity to the original structures. This indicates that the model is not performing rational edits; rather, it generates different but valid molecules that happen to satisfy the criteria. In other words, BioT5-base relies more on recalling patterns from its training corpus than on learning to systematically modify molecular structures. Consequently, while the ChEBI-20 dataset is valuable for molecule–caption translation, its limited diversity, scale, and one-to-one mapping nature are insufficient for training LLMs to perform nuanced, open-ended molecular generation.

**F4: The data scaling law does not always hold: task-specific bottlenecks limit gains in molecule optimization and editing.** As shown in Figure 3, our experiments with Galactica-125M across the five data scales provide new insights into the data scaling law for molecule generation. Overall, we observe that larger datasets generally improve performance, but the effect is highly task-dependent.

In **MolCustom**, scaling the dataset from large to xlarge yields a remarkable 286% average performance gain, indicating that tasks involving complex, de novo synthesis strongly benefit from larger and more diverse training data. By contrast, Galactica-125M obtains only modest gains in **MolOpt**, and even negative gains in **MolEdit**, suggesting that additional data alone cannot overcome existing bottlenecks in these relatively simpler tasks.

Interestingly, when examining Llama-3.1-8B-Instruct, we find that performance continues to increase when scaling from the large to xlarge data level, except for LogP and MR subtasks. This contrast generally highlights that small models like Galactica-125M are capacity-limited, whereas larger models can more effectively leverage additional data.

To summarize, these findings suggest that while complex, open-ended tasks are data-driven, simpler property optimization or editing tasks may be constrained more by model capacity than by dataset size. Thus, merely enlarging training data may not yield proportional benefits. Instead, future progress will require a balanced consideration of both model scale and data scale.

## 5 CONCLUSION

In this study, we introduced **$S^2$-Bench** and **OpenMolIns**, the **first** benchmark and instruction tuning dataset for evaluating the capabilities of LLMs in open-domain natural language-driven molecule generation. By moving beyond traditional one-to-one text-to-molecule mappings, our benchmark accommodates the open-ended nature of molecular discovery. $S^2$-Bench focuses on the realistic molecular reasoning and design, rather than simple pattern matching and memorization. Our comprehensive benchmarking of 30 LLMs not only highlights the challenges faced by current models and the limitations of existing targeted molecule generation datasets, but also demonstrates the substantial potential of LLMs for natural language–driven molecule discovery.

ETHICS STATEMENT

This work does not involve ethic issues.

REPRODICIBILITY STATEMENT

We have attached all the test examples in our benchmark in the supplementary materials.

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

## A    IMPLEMENTATION DETAILS

We implement various scripts to facilitate the testing of the aforementioned models. For proprietary models, we adopt the OpenAI API[1] framework. For open-source general LLMs, we utilize both the VLLM[2] framework and the OpenAI framework. For the remaining LLMs, we adopt the Hugging Face transformers library[3] for inference.

Furthermore, BioT5 is designed to use SELFIES, an alternative string representation of molecules by using special tokens to ensure valid generations (Krenn et al., 2019), as input instead of SMILES. Consequently, we convert the molecule SMILES strings into SELFIES format on BioT5.

## B    HYPER PARAMETERS

In this section, we illustrate the detailed parameters adopted in this work, as shown in Table 4. Notably, we utilize one NVIDIA A100 80G for testing open-source LLMs, and $4\times$NVIDIA A100 80G for instruction tuning. For close-source LLMs, we call the official APIs of them.

| Item | Value |
|------|-------|
| *Generation* | |
| temperature | 0.75 |
| top_p | 0.85 |
| num_beams | 1 |
| max_new_tokens | 512 |
| *Instruction Tuning* | |
| epochs(light) | 10 |
| epochs(small, medium) | 5 |
| epochs(large, xlarge) | 3 |
| batchsize | 32 |
| lr | 3e-4 |
| warmup_ratio | 0.03 |
| cutoff_len | 1024 |
| *Lora Settings* | |
| r | 64 |
| $\alpha$ | 128 |
| dropout | 0.1 |

Table 4: Hyper-parameters.

## C    DATA CONSTRUCTION

In this section, we introduce the construction details of $S^2$-Bench and OpenMolIns dataset, as well as the prompt templates. Figure 4 demonstrates the overall data construction process for $S^2$-Bench as well as the OpenMolIns. Generally, we utilize Zinc250K to construct the test samples in $S^2$-Bench, while adopting PubChem for the training samples in OpenMolIns. Notably, when constructing the OpenMolIns, we exclude all the molecules in Zinc250K to avoid data leakage and ensure the novelty score of the generated molecules.

Zinc-250K is widely adopted in molecular generation research due to its well-curated molecular structures, and balanced coverage of drug-like chemical space, which ensures fair and reproducible evaluation. Here, we report molecular distribution statistics for the Zinc-250K, PubChem, and our pre-selected samples used in MolEdit and MolOpt. Specifically, we compare the average (i) atom count, (ii) ring count, (iii) branch count, and (iv) path length across datasets.

As shown in Table 5, Zinc-250K exhibits broadly similar statistical trends to PubChem, suggesting that Zinc-250K captures the core characteristics of drug-like chemistry and remains a reasonable proxy for discovery-scale small molecules. Meanwhile, the distribution of our pre-selected samples aligns closely with the original Zinc-250K, suggesting that our pre-selection does not substantially distort the underlying molecular distribution.

---

[1]https://platform.openai.com/docs/

[2]https://docs.vllm.ai

[3]https://huggingface.co/docs/

| Dataset | # Samples | Avg Atom Count | Avg Ring Count | Avg Branch Count | Avg Path Length |
|---|---|---|---|---|---|
| Zinc-250K | 250,000 | 23.15 | 2.76 | 7.31 | 12.48 |
| PubChem | 9,000,000 | 25.18 | 2.8 | 7.87 | 13.07 |
| S²-Bench's sampling | 30,000 | 23.17 | 2.74 | 7.34 | 12.47 |

Table 5: Data Distribution Statistics.

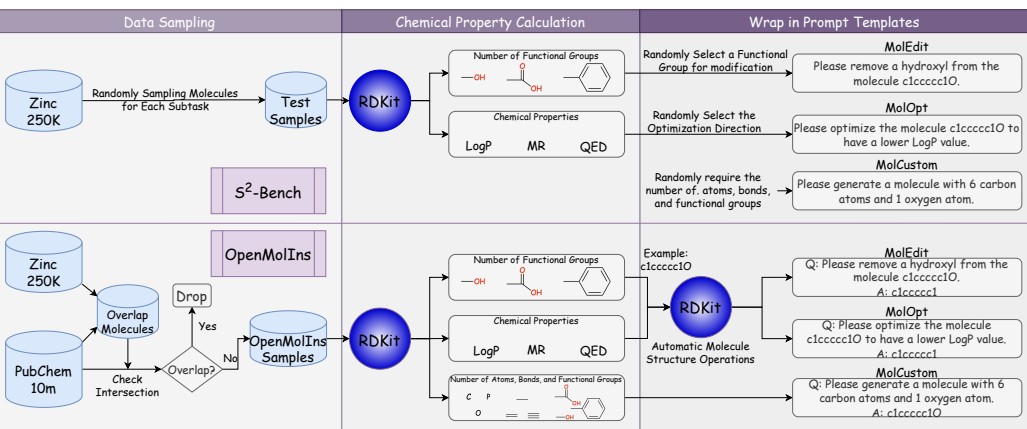

Figure 4: Data construction workflow of S$^2$-Bench & OpenMolIns.

## C.1 MOLEDIT

For the MolEdit task, we consider the common operations on modifying functional groups in a given molecule (i.e., add, drop, and substitute), which are simple tasks for human experts but challenging to LLMs. In this case, we further develop three corresponding subtasks: *AddComponent*, *DelComponent*, and *SubComponent*. Prompt templates for MolEdit are shown in Table 6. However, there are different kinds of functional groups, and some functional groups can play an important role in the molecule structure, such as connecting two separate parts of the molecule, which makes them unsuitable for these operations above as these operations will entirely change the structure of the molecule. In this case, we aim to make a slight change in the molecule structure and limit most of the functional groups we choose within the end groups.

| **Prompt Templates for MolEdit** |
|---|
| *AddComponent* |
| Please add a {} to the molecule {}. |
| Modify the molecule {} by adding a {}. |
| Add a {} to the molecule {}. |
| *DelComponent* |
| Please remove a {} from the molecule {}. |
| Modify the molecule {} by removing a {}. |
| Remove a {} from the molecule {}. |
| *SubComponent* |
| Please substitute a {} in the molecule {} by {}. |
| Modify the molecule {} by replacing a {} by {}. |
| Replace a {} in the molecule {} by {}. |
| Please replace a {} in the molecule {} with {}. |
| Modify the molecule {} by substituting a {} with {}. |
| Substitute a {} in the molecule {} with {}. |

Table 6: Prompt Templates for MolEdit.

Table 7 presents the functional groups that are taken into account for *AddComponent* and *DelComponent*, along with their respective selection weights. Given that certain functional groups are encountered less frequently in real-world scenarios, we have implemented a weighted random selection process for *AddComponent*, which ensures that less common functional groups are assigned a lower probability to be chosen.

| Functional Group | benzene ring | hydroxyl | aldehyde | carboxyl | amide |
|---|---|---|---|---|---|
| Weights | 15 | 15 | 5 | 5 | 10 |

| Functional Group | amine | nitro | halo | nitrile | thiol |
|---|---|---|---|---|---|
| Weights | 5 | 5 | 5 | 1 | 1 |

Table 7: Functional Groups that are considered in *AddComponent* and *DelComponent*, as well as their weights to be selected in *AddComponent*.

For *SubComponent*, our focus is exclusively on end groups for simplification, which include hydroxyl, aldehyde, carboxyl, nitro, halo, nitrile, and thiol, which ensures that the editing operations are confined to substituting the existing functional group with another from this list, thereby maintaining the integrity of the molecule's overall structure without altering it fundamentally.

## C.2 MOLOPT

MolOpt is designed to optimize molecular properties through the refinement of molecule structures, is not a brand-new task. Previously, GNN-based methods have been widely adopted in this task, while these methods can only help with one specific subtask at a time. In contrast, $S^2$-Bench requires one single LLM to optimize molecules with different metrics and optimization directions. In this work, we specifically focus on enhancing specific characteristics that are crucial for drug discovery and chemical synthesis, including LogP, MR, and QED. The prompt templates for MolOpt are listed in Table 8.

| **Prompt Templates for MolOpt** |
|---|
| *LogP* |
| Please optimize the molecule {} to have a lower/higher LogP value. |
| Modify the molecule {} to decrease/increase its LogP value. |
| Optimize the molecule {} to have a lower/higher LogP value. |
| Please modify the molecule {} to decrease/increase its LogP value. |
| Modify the molecule {} to have a lower/higher LogP value. |
| *MR* |
| Please optimize the molecule {} to have a lower/higher MR value. |
| Modify the molecule {} to decrease/increase its MR value. |
| Optimize the molecule {} to have a lower/higher MR value. |
| Please modify the molecule {} to decrease/increase its MR value. |
| Modify the molecule {} to have a lower/higher MR value. |
| *QED* |
| Please optimize the molecule {} to have a lower/higher QED value. |
| Modify the molecule {} to decrease/increase its QED value. |
| Optimize the molecule {} to have a lower/higher QED value. |
| Please modify the molecule {} to decrease/increase its QED value. |
| Modify the molecule {} to have a lower/higher QED value. |

Table 8: Prompt Templates for MolOpt.

Below are the detailed explanations of the chemical properties adopted in the three subtasks of MolOpt:

*LogP* refers to the logarithm of the partition coefficient, which is a measure of a molecule's hydrophilicity or lipophilicity. It is an important factor in determining a compound's bioavailability and membrane permeability.

*Molecular Refractivity (MR)* is a measure of the molar refractive index, which provides insight into the molecular size and the degree of molecular branching. It is used to assess the overall shape and bulk of a molecule.

*Quantitative Estimation of Drug-Likeness (QED)* is a computational metric that evaluates the drug-likeness of a molecule based on a set of predefined rules. A higher QED score suggests a greater likelihood that the molecule will have favourable pharmacological properties.

Among the three subtasks, the QED optimization, which assesses the potential of a molecule to become a drug for curing diseases, is generally considered the most challenging property for molecule optimization. In contrast, LogP and MR can be more directly inferred from the molecule's structure, thus making them more straightforward to optimize.

## C.3  MolCustom

To enable customized design of molecules, we consider three fundamental features for describing the molecule, including atoms, bonds, and functional groups. Given the specified category and number of atoms, bonds, and functional groups, LLMs should generate the molecule as we request. The prompt templates for MolCustom are shown in Table 9. Below, we present the construction details of the three subtasks for MolCustom:

---

**Prompt Templates for MolCustom**

*AtomNum*
Please generate a molecule with {} atom(s).
Please generate a molecule composed of {} atom(s).
Please generate a molecule consisting {} atom(s).
The molecule has {} atom(s).
The molecule is composed of {} atom(s).
The molecule consists of {} atom(s).
There is a molecule with {} atom(s).
There is a molecule composed of {} atom(s).
There is a molecule consisting of {} atom(s).
The molecule contains {} atom(s).

*BondNum*
Please generate a molecule with {} bond(s).
Please generate a molecule composed of {} bond(s).
Please generate a molecule consisting {} bond(s).
The molecule has {} bond(s).
The molecule is composed of {} bond(s).
The molecule consists of {} bond(s).
There is a molecule with {} bond(s).
There is a molecule composed of {} bond(s).
There is a molecule consisting of {} bond(s).
The molecule contains {} bond(s).

*FunctionalGroup*
Please generate a molecule with {} group(s).
Please generate a molecule composed of {} group(s).
Please generate a molecule consisting {} group(s).
The molecule has {} group(s).
The molecule is composed of {} group(s).
The molecule consists of {} group(s).
There is a molecule with {} group(s).
There is a molecule composed of {} group(s).
There is a molecule consisting of {} group(s).
The molecule contains {} group(s).

---

Table 9: Prompt Templates for MolCustom.

| Atom | carbon | oxygen | nitrogen | sulfur | fluorine | chlorine | bromine | iodine | phosphorus |
|------|--------|--------|----------|--------|----------|----------|---------|--------|------------|
| Weights | [Mandatory] | 5 | 3 | 3 | 2 | 2 | 2 | 2 | 1 |

| Atom | boron | silicon | selenium | tellurium | arsenic | antimony | bismuth | polonium |
|------|-------|---------|----------|-----------|---------|----------|---------|----------|
| Weights | 1 | 1 | 1 | 1 | 1 | 1 | 1 | 1 |

Table 10: Atoms that are considered in *AtomNum*, as well as their weights to be selected. Higher weights indicate that they are more likely to be selected. For simplification, we only consider normal atoms in the molecule structures.

*AtomNum.* Table 10 shows the atoms we consider in *AtomNum*, as well as their weights to be selected. Notably, carbon, as the basic unit in organic chemicals, is a mandatory option. The number of carbon

atoms ranges from 1 to 40, while the number of other selected atoms ranges from 1 to 5. This setting relieves the difficulty for generation, as LLMs could generate a carbon backbone first and attach the remaining atoms to the backbone one by one.

| Bond | single | double | triple | rotatable | aromatic |
|---|---|---|---|---|---|
| Weights | 5 | 4 | 3 | 1 | 1 |

Table 11: Chemical bonds that are considered in *BondNum*, as well as their weights to be selected. Higher weights indicate that they are more likely to be selected. For simplification, we only consider normal bonds in the molecule structures.

*BondNum.* Similarly, we select five different kinds of chemical bonds: single, double, triple, rotatable, and aromatic, as shown in Table 11. For the single bond, if selected, the number can vary from 1 to 50. For the aromatic bond, the number follows the rules of the formation of aromatic bonds, varying from 5 to 20. Moreover, the number of these remaining bonds, if selected, is specified from 1 to 5.

| Functional Group | benzene ring | hydroxyl | anhydride | aldehyde | ketone | carboxyl | ester | amide | amine | nitro |
|---|---|---|---|---|---|---|---|---|---|---|
| Weights | 15 | 15 | 2 | 5 | 5 | 10 | 5 | 5 | 5 | 2 |

| Functional Group | halo | thioether | nitrile | thiol | sulfide | disulfide | sulfoxide | sulfone | borane |
|---|---|---|---|---|---|---|---|---|---|
| Weights | 2 | 1 | 1 | 1 | 1 | 1 | 1 | 1 | 1 |

Table 12: Functional Groups that are considered in *FunctionalGroup*, as well as their weights to be selected. Higher weights indicate that they are more likely to be selected. For simplification, we only consider normal and edge functional groups in the molecule structures.

*FunctionalGroup.* Lastly, we also specify functional groups in the molecule structure. Table 12 shows the range of functional groups and their weights that are taken into consideration.

Notably, in MolCustom, if not required, LLMs can generate any number of these unspecified atoms, bonds, and functional groups. However, for these specified items, LLMs should strictly follow the requirements.

# D    TESTING PROCESS

In this section, we present the testing process used to calculate the success rate of each subtask.

---

**Algorithm 1** MolEdit Testing Process

---

**Input:** Generated Molecule $m^g$, Original Molecule $m^o$, Subtask $t$, Group-to-add $\alpha$, Group-to-delete $\Delta$

**Output:** Pass or Not (**Bool**)

    **if** $t$ is AddComponent **then**

        **if** $|\alpha|(m^g) = |\alpha|(m^o) + 1$ **then**

            return **true**

        **else**

            return **false**

    **else if** $t$ is DelComponent **then**

        **if** $|\Delta|(m^g) = |\Delta|(m^o)$ - 1 **then**

            return **true**

        **else**

            return **false**

    **else if** $t$ is SubComponent **then**

        **if** $|\Delta|(m^g) = |\Delta|(m^o)$ - 1 **and** $|\alpha|(m^g) = |\alpha|(m^o) + 1$ **then**

            return **true**

        **else**

            return **false**

---

---

**Algorithm 2** MolOpt Testing Process

---

**Input:** Generated Molecule $m^g$, Original Molecule $m^o$, Subtask $t$, Optimization Direction Requirement $R$

**Output:** Pass or Not (**Bool**)

    **if** $t$ is LogP **then**
        **if** $LogP(m^g) > LogP(m^o)$ **and** R is $\uparrow$ **then**
            return **true**
        **else if** $LogP(m^g) < LogP(m^o)$ **and** R is $\downarrow$ **then**
            return **true**
        **else**
            return **false**
    **else if** $t$ is MR **then**
        **if** $MR(m^g) > MR(m^o)$ **and** R is $\uparrow$ **then**
            return **true**
        **else if** $MR(m^g) < MR(m^o)$ **and** R is $\downarrow$ **then**
            return **true**
        **else**
            return **false**
    **else if** $t$ is QED **then**
        **if** $QED(m^g) > QED(m^o)$ **and** R is $\uparrow$ **then**
            return **true**
        **else if** $QED(m^g) < QED(m^o)$ **and** R is $\downarrow$ **then**
            return **true**
        **else**
            return **false**

---

The testing process for **MolEdit** task is demonstrated in Algorithm 1. However, in this process, we can not guarantee that the modified molecule differs the original molecule only in the specified functional group. The reasons are to follow:

- The complexity of molecular structures makes it difficult to use regular matching to monitor the exact number of functional groups present. This is because molecules can have a wide variety of bonding patterns and spatial arrangements that are not always amenable to simple, rule-based matching techniques.

- Modifying functional groups in specific positions can indeed lead to unintended changes in the molecule's overall structure. This is due to the interconnected nature of atoms within a molecule, where altering one part can have a ripple effect on the stability and geometry of the entire structure.

- The tools and algorithms currently available may not have the necessary sophistication to predict and control all the possible outcomes of a functional group modification, particularly when dealing with complex molecules or reactions that are not fully understood.

Therefore, the similarity score needs to be considered to ensure the accuracy and quality of the generation.

For **MolOpt**, we adopt Algorithm 2 for the testing process. Similarly, merely passing the test does not necessarily mean that the generated molecule follows the principle of minimal change.

For **MolCustom**, the testing process is shown in Algorithm 3, which test whether the specified number of the atom, bond, or functional group satisfies the requirements.

## E    JUSTIFICATION FOR SIMILARITY TERMS IN WEIGHTED SUCCESS RATE

In this section, we conducted an extra correlation analysis to justify our design for using similarity in Weighted Success Rate. Here, we focused on the unusual performance of BioT5-base in MolEdit and MolOpt, where the model achieves high success rates but performs poorly in the quality term, indicating that the generated molecules are not modified from the original molecule but from scratch.

---

**Algorithm 3** MolCustom Testing Process

---

**Input:** Generated Molecule $m^g$, Subtask $t$, Atom List $A$, Bond List $B$, Functional Group List $G$, Requirements $R$
**Output:** Pass or Not (**Bool**)
  flag = **true**
  **if** $t$ is AtomNum **then**
    **for** $atom$ in $A$ **do**
      **if** $|atom|(m^g) \neq R[atom]$ **then**
        flag = **false**
  **else if** $t$ is BondNum **then**
    **for** $bond$ in $B$ **do**
      **if** $|bond|(m^g) \neq R[bond]$ **then**
        flag = **false**
  **else if** $t$ is FunctionalGroup **then**
    **for** $group$ in $G$ **do**
      **if** $|group|(m^g) \neq R[group]$ **then**
        flag = **false**
  return flag

---

For this analysis, we first randomly sampled 100 examples from each subtask and then asked human experts to evaluate the correctness and quality of the generated molecules. Each example is assigned an expert score from 0 to 5, where:

- 5 denotes a perfect success,

- 4 means success but with redundant operations (a little different structure),

- 3 shows failure but modified from the original molecule (similar structure),

- 2 denotes failure and a little different structure,

- 1 indicates success but an unrelated/different structure,

- 0 denotes failure and an unrelated/different structure.

We calculate the average score of expert evaluation. The results are shown in Table 13.

| BioT5-base | Subtasks | SR | Similarity | WSR | Validity | Human Expert Evaluation |
|---|---|---|---|---|---|---|
| MolEdit | AddComponent | 0.3462 | 0.1567 | 0.0542 | 1 | 0.64 |
| | DelComponent | 0.1668 | 0.1597 | 0.0266 | 1 | 0.22 |
| | SubComponent | 0.0684 | 0.1576 | 0.0108 | 0.9998 | 0.19 |
| MolOpt | LogP | 0.5158 | 0.1526 | 0.0787 | 1 | 0.58 |
| | MR | 0.506 | 0.1597 | 0.0808 | 1 | 0.67 |
| | QED | 0.5068 | 0.158 | 0.0801 | 1 | 0.64 |

Table 13: Human Expert Judgements for WSR. Here, we focused on the unusual performance of BioT5-base, where the model achieves high success rates but performs poorly in the quality term.

We computed both Pearson and Spearman correlations between SR/WSR and the expert evaluation scores across the subtasks. The results show that WSR exhibits consistently stronger alignment with expert ratings. Specifically, WSR achieves a slightly higher Pearson correlation with expert scores (r = 0.926 vs. 0.922 for SR) and, more importantly, a substantially higher Spearman rank correlation ($\rho$ = 0.899 vs. 0.551 for SR). While the correlation between SR and expert ratings is not statistically significant (p = 0.257), WSR shows strong and significant rank consistency (p = 0.015).

Therefore, it is essential to apply similarities as weights of Success Rate to reflect the genuine capabilities of LLMs more faithfully.

# F  DETAILED RESULTS

In this section, we first show the leaderboard of S²-Bench in Table 14, where Claude-3.5 achieves the second place with an average weighted success rate of 35.92%. Notably, via instruction tuning on our OpenMolIns dataset, Llama3.1-8B achieves the first place, which outperforms all the other LLMs.

| Model | #Parameters (B) | $\overline{SR}$ (%) | $\overline{WSR}$(%) | Rank |
|---|---|---|---|---|
| Llama3.1-8B (OpenMolIns-xlarge) | 8 | 58.79 | 39.33 | 1 |
| Claude-3.5 (Anthropic, 2024b) | - | 51.10 | 35.92 | 2 |
| Gemini-1.5-pro (Deepmind, 2024) | - | 52.25 | 34.80 | 3 |
| GPT-4-turbo (OpenAI, 2023) | - | 50.74 | 34.23 | 4 |
| GPT-4o (OpenAI, 2023) | - | 49.08 | 32.29 | 5 |
| Claude-3 (Anthropic, 2024a) | - | 46.14 | 30.47 | 6 |
| Llama3.1-8B (OpenMolIns-large) | 8 | 43.1 | 27.22 | 7 |
| Galactica-125M (OpenMolIns-xlarge) | 0.125 | 44.48 | 25.73 | 8 |
| Llama3-70B-Instruct (Int4) (Dubey et al., 2024) | 70 | 38.54 | 23.93 | 9 |
| Galactica-125M (OpenMolIns-large) | 0.125 | 39.28 | 23.42 | 10 |
| Galactica-125M (OpenMolIns-medium) | 0.125 | 34.54 | 19.89 | 11 |
| GPT-3.5-turbo (OpenAI, 2023) | - | 28.93 | 18.58 | 12 |
| Galactica-125M (OpenMolIns-small) | 0.125 | 24.17 | 15.18 | 13 |
| Gemma3-12B (Team et al., 2025) | 12 | 26.28 | 15.00 | 14 |
| Deepseek-R1-distill-Qwen-7B (Guo et al., 2025) | 7 | 25.07 | 14.61 | 15 |
| Llama3.1-8B-Instruct (Dubey et al., 2024) | 8 | 26.26 | 14.09 | 16 |
| Llama3-8B-Instruct (Dubey et al., 2024) | 8 | 26.40 | 13.75 | 17 |
| chatglm-9B (GLM et al., 2024) | 9 | 18.50 | 13.13(7) | 18 |
| Galactica-125M (OpenMolIns-light) | 0.125 | 20.95 | 13.13(6) | 19 |
| ChemDFM-v1.5-8B (Zhao et al., 2025) | 8 | 18.24 | 12.07 | 20 |
| ChemLLM-20B (Zhang et al., 2024) | 20 | 16.23 | 9.76 | 21 |
| Llama3.2-1B (OpenMolIns-large) | 1 | 14.11 | 8.10 | 22 |
| yi-1.5-9B (Young et al., 2024) | 9 | 14.10 | 7.32 | 23 |
| Mistral-7B-Instruct-v0.2 (Jiang et al., 2023) | 7 | 11.17 | 4.81 | 24 |
| BioT5-base (Pei et al., 2023) | 0.25 | 24.19 | 4.21 | 25 |
| MolT5-large (Edwards et al., 2022) | 0.78 | 23.11 | 2.89 | 26 |
| Llama3.1-1B-Instruct (Dubey et al., 2024) | 1 | 3.95 | 1.99 | 27 |
| MolT5-base (Edwards et al., 2022) | 0.25 | 11.11 | 1.30(0) | 28 |
| MolT5-small (Edwards et al., 2022) | 0.08 | 11.55 | 1.29(9) | 29 |
| Qwen2-7B-Instruct (Yang et al., 2024) | 7 | 0.18 | 0.15 | 30 |

Table 14: Leaderboard of S²-Bench.

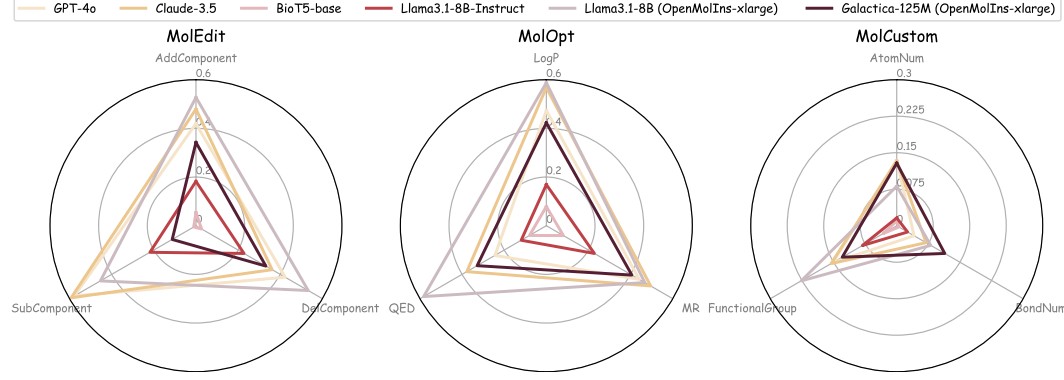

Figure 5: Performance Comparison across several representative LLMs on S²-Bench.

We also compare several representative LLMs from different model categories via radar maps, which are shown in Figure 5. Among the proprietary models, it can be concluded that Claude-3.5 achieves a

powerful performance in these subtasks, except for the DelComponent task and the BondNum task. In DelComponent task, GPT-4o demonstrates better performance than Claude-3.5. More importantly, we can also observe that Llama3.1-8B (OpenMolIns-xlarge) outperforms most of the other LLMs across these nine subtasks, achieving the SOTA performance.

Meanwhile, these radar maps also imply the difficulties of the subtasks. For example, for majority of the LLMs, they are struggling with the BondNum task. Even the most advanced proprietary LLMs like Claude-3.5 failed to achieve an ideal performance.

Finally, we also include a reasoning LLM, Deepseek-R1-distill-Qwen-7B (Guo et al., 2025), as shown in Table 14, which achieves the best performance among 7 and 8B level open-source general LLMs, showing the potential of LLM reasoning in open-domain natural language-driven molecule generation tasks.

Finally, we present the detailed experimental results of all the subtasks in Table 15, 16, and 17. Note that **we do not provide results of vanilla galactica-125M, for it's not instruction tuned and cannot respond to human instructions.** All results of vanilla galactica-125M should be close to zeros.

| Models | AddComponent | | | | DelComponent | | | | SubComponent | | | |
|---|---|---|---|---|---|---|---|---|---|---|---|---|
| | SR | Similarity | WSR | Validity | SR | Similarity | WSR | Validity | SR | Similarity | WSR | Validity |
| GPT-4o (OpenAI, 2023) | 0.6188 | 0.6782 | 0.4197 | 0.7412 | 0.7012 | 0.6038 | 0.4234 | 0.8474 | 0.7992 | 0.7225 | 0.5774 | 0.9368 |
| GPT-4-turbo (OpenAI, 2023) | 0.6990 | 0.6936 | 0.4848 | 0.7934 | 0.7244 | 0.5735 | 0.4154 | 0.9060 | 0.7778 | 0.7323 | 0.5696 | 0.9160 |
| GPT-3.5-turbo (OpenAI, 2023) | 0.5832 | 0.6545 | 0.3817 | 0.7980 | 0.3082 | 0.7797 | 0.2403 | 0.8468 | 0.2918 | 0.6333 | 0.1848 | 0.6822 |
| Claude-3.5 (Anthropic, 2024b) | 0.6832 | 0.7017 | 0.4794 | 0.4414 | 0.5414 | 0.6678 | 0.3615 | 0.7960 | 0.8104 | 0.7310 | 0.5924 | 0.9588 |
| Claude-3 (Anthropic, 2024a) | 0.6766 | 0.6840 | 0.4628 | 0.8180 | 0.5556 | 0.6408 | 0.3560 | 0.8984 | 0.6550 | 0.7159 | 0.4689 | 0.9184 |
| Gemini-1.5-pro (Deepmind, 2024) | 0.7058 | 0.6792 | 0.4794 | 0.8254 | 0.7590 | 0.5949 | 0.4515 | 0.9158 | 0.7148 | 0.7139 | 0.5103 | 0.8684 |
| Llama3-70B-Instruct (Int4) (Dubey et al., 2024) | 0.5198 | 0.6801 | 0.3535 | 0.5922 | 0.6122 | 0.5637 | 0.3451 | 0.7182 | 0.5094 | 0.7170 | 0.3652 | 0.6822 |
| Llama3-8B-Instruct (Dubey et al., 2024) | 0.3914 | 0.6649 | 0.2602 | 0.5374 | 0.4348 | 0.5058 | 0.2199 | 0.5700 | 0.2602 | 0.6841 | 0.1780 | 0.4838 |
| Llama3.1-8B-Instruct (Dubey et al., 2024) | 0.2992 | 0.6088 | 0.1822 | 0.4962 | 0.4336 | 0.5257 | 0.2279 | 0.5910 | 0.3401 | 0.6424 | 0.2185 | 0.5076 |
| Mistral-7B-Instruct-v0.2 (Jiang et al., 2023) | 0.1868 | 0.6251 | 0.1168 | 0.3760 | 0.2018 | 0.3774 | 0.0762 | 0.3590 | 0.0602 | 0.6227 | 0.0375 | 0.3550 |
| Qwen2-7B-Instruct (Yang et al., 2024) | 0.0010 | 0.2527 | 0.0003 | 0.0036 | 0.0006 | 0.4024 | 0.0002 | 0.0012 | 0.0004 | 0.2895 | 0.0001 | 0.0068 |
| Yi-1.5-9B (Young et al., 2024) | 0.1742 | 0.4170 | 0.0726 | 0.4216 | 0.2858 | 0.5936 | 0.1697 | 0.4909 | 0.1370 | 0.4619 | 0.0633 | 0.4368 |
| Chatglm-9B (GLM et al., 2024) | 0.2932 | 0.7622 | 0.2235 | 0.5686 | 0.2956 | 0.7494 | 0.2215 | 0.6914 | 0.1498 | 0.7150 | 0.1071 | 0.5084 |
| Llama3.2-1B-Instruct (Dubey et al., 2024) | 0.0374 | 0.5343 | 0.0200 | 0.1982 | 0.0768 | 0.5750 | 0.0442 | 0.3028 | 0.0102 | 0.3671 | 0.0037 | 0.1468 |
| Gemma3-12B (Team et al., 2025) | 0.4300 | 0.7081 | 0.3045 | 0.6324 | 0.4948 | 0.5158 | 0.2552 | 0.7168 | 0.4208 | 0.7154 | 0.3010 | 0.7616 |
| Deepseek-R1-distill-Qwen-7B (Guo et al., 2025) | 0.2667 | 0.6904 | 0.1841 | 0.0090 | 0.6367 | 0.5921 | 0.3770 | 0.0356 | 0.2156 | 0.7380 | 0.1591 | 0.0356 |
| ChemDFM-v1.5-8B (Zhao et al., 2025) | 0.2942 | 0.6454 | 0.1899 | 0.9112 | 0.2056 | 0.6937 | 0.1426 | 0.9286 | 0.2198 | 0.6432 | 0.1414 | 0.8652 |
| ChemLLM-20B (Zhang et al., 2024) | 0.1436 | 0.5978 | 0.0858 | 0.2108 | 0.2092 | 0.5734 | 0.1200 | 0.3098 | 0.3088 | 0.6805 | 0.2101 | 0.4936 |
| MolT5-small (Edwards et al., 2022) | 0.1220 | 0.1027 | 0.0125 | 0.4490 | 0.1598 | 0.1125 | 0.0180 | 0.4504 | 0.0708 | 0.1029 | 0.0073 | 0.4876 |
| MolT5-base (Edwards et al., 2022) | 0.1354 | 0.1066 | 0.0144 | 0.4686 | 0.1562 | 0.1144 | 0.0179 | 0.4472 | 0.0584 | 0.1028 | 0.0060 | 0.4426 |
| MolT5-large (Edwards et al., 2022) | 0.2834 | 0.1084 | 0.0307 | 0.9282 | 0.2228 | 0.1201 | 0.0268 | 0.9198 | 0.1692 | 0.0932 | 0.0158 | 0.9410 |
| BioT5-base (Pei et al., 2023) | 0.3462 | 0.1567 | 0.0542 | 1.0000 | 0.1668 | 0.1597 | 0.0266 | 1.0000 | 0.0684 | 0.1576 | 0.0108 | 0.9998 |
| Llama3.2-1B (OpenMolIns-large) | 0.1756 | 0.5676 | 0.0997 | 0.3216 | 0.1816 | 0.4963 | 0.0901 | 0.2466 | 0.0844 | 0.5415 | 0.0457 | 0.2958 |
| Llama3.1-8B (OpenMolIns-large) | 0.5822 | 0.6541 | 0.3808 | 0.6730 | 0.5104 | 0.5074 | 0.2590 | 0.6896 | 0.5440 | 0.6258 | 0.3404 | 0.8400 |
| Llama3.1-8B (OpenMolIns-xlarge) | 0.7790 | 0.6769 | 0.5273 | 0.9468 | 0.8640 | 0.6166 | 0.5327 | 0.9132 | 0.6100 | 0.7434 | 0.4535 | 0.9596 |
| Galactica-125M (OpenMolIns-light) | 0.3786 | 0.5958 | 0.2256 | 0.6842 | 0.2062 | 0.6521 | 0.1345 | 0.7048 | 0.3102 | 0.5879 | 0.1824 | 0.6674 |
| Galactica-125M (OpenMolIns-small) | 0.3472 | 0.6172 | 0.2143 | 0.5356 | 0.3258 | 0.6025 | 0.1963 | 0.5758 | 0.2692 | 0.6181 | 0.1664 | 0.5692 |
| Galactica-125M (OpenMolIns-medium) | 0.4736 | 0.5682 | 0.2691 | 0.7442 | 0.4886 | 0.5184 | 0.2533 | 0.7488 | 0.3282 | 0.5975 | 0.1961 | 0.6958 |
| Galactica-125M (OpenMolIns-large) | 0.5866 | 0.5876 | 0.3447 | 0.8228 | 0.6078 | 0.5577 | 0.3390 | 0.7934 | 0.3438 | 0.6491 | 0.2232 | 0.8438 |
| Galactica-125M (OpenMolIns-xlarge) | 0.5842 | 0.5859 | 0.3423 | 0.8438 | 0.6526 | 0.5084 | 0.3318 | 0.8286 | 0.1872 | 0.6024 | 0.1128 | 0.8538 |

Table 15: Detailed results on MolEdit. For each task, we highlight the best and the second-best success rate (SR), as well as the weighted success rate (WSR).

# G  LIMITATIONS

Although S$^2$-Bench is carefully designed and well-validated through our experiments, we still observe several limitations:

**Data Distribution.** In our data construction process, we allocate distributions to atoms, bonds, and functional groups with the aim of making our benchmark more reflective of real-world distributions. Nevertheless, the distribution we use is largely empirical and may not be sufficiently accurate to reconstruct real-world scenarios accurately.

**The Selection of Subtasks.** For the selection of subtasks, there are various choices in MolOpt, we compeletly understand that there are multiple viable options, especially when there are enormous metrics and real-world experimental datasets for evaluating molecular properties. However, we need to 1) consider the consistency between the three major tasks in S$^2$-Bench and balance their weights (which means we can not select too many metrics for MolOpt, otherwise it would make the benchmark solely an optimization benchmark), and 2) ensure that the metrics can be efficiently and robustly calculated. Therefore, we only select the three common metrics, LogP, MR, and QRD, for MolOpt.

| Models | LogP | | | | MR | | | | QED | | | |
|---|---|---|---|---|---|---|---|---|---|---|---|---|
| | SR | Similarity | WSR | Validity | SR | Similarity | WSR | Validity | SR | Similarity | WSR | Validity |
| GPT-4o (OpenAI, 2023) | 0.7190 | 0.6586 | 0.4735 | 0.8796 | 0.6864 | 0.6420 | 0.4407 | 0.8352 | 0.3952 | 0.6180 | 0.2442 | 0.8570 |
| GPT-4-turbo (OpenAI, 2023) | 0.7662 | 0.6984 | 0.5351 | 0.9048 | 0.7388 | 0.6821 | 0.5039 | 0.8848 | 0.3946 | 0.6587 | 0.2599 | 0.9050 |
| GPT-3.5-turbo (OpenAI, 2023) | 0.4048 | 0.6327 | 0.2561 | 0.8540 | 0.4120 | 0.6263 | 0.2580 | 0.8486 | 0.3316 | 0.5635 | 0.1869 | 0.8354 |
| Claude-3.5 (Anthropic, 2024b) | 0.7970 | 0.7124 | 0.5678 | 0.9422 | 0.6962 | 0.7112 | 0.4951 | 0.9110 | 0.5361 | 0.7042 | 0.3775 | 0.8604 |
| Claude-3 (Anthropic, 2024a) | 0.7984 | 0.6067 | 0.4844 | 0.9096 | 0.6094 | 0.6398 | 0.3899 | 0.9062 | 0.4678 | 0.5855 | 0.2739 | 0.9044 |
| Gemini-1.5-pro (Deepmind, 2024) | 0.7712 | 0.7022 | 0.5415 | 0.9274 | 0.7876 | 0.6744 | 0.5312 | 0.8926 | 0.4704 | 0.6077 | 0.2859 | 0.9484 |
| Llama3-70B-Instruct (Int4) (Dubey et al., 2024) | 0.5984 | 0.6028 | 0.3607 | 0.6482 | 0.5684 | 0.6032 | 0.3429 | 0.6272 | 0.2774 | 0.4828 | 0.1339 | 0.6340 |
| Llama3-8B-Instruct (Dubey et al., 2024) | 0.4642 | 0.3658 | 0.1698 | 0.6086 | 0.4332 | 0.4793 | 0.2076 | 0.5704 | 0.2568 | 0.4547 | 0.1168 | 0.6112 |
| Llama3.1-8B-Instruct (Dubey et al., 2024) | 0.3990 | 0.4235 | 0.1690 | 0.5122 | 0.4336 | 0.5257 | 0.2279 | 0.5910 | 0.2655 | 0.4499 | 0.1194 | 0.6158 |
| Mistral-7B-Instruct-v0.2 (Jiang et al., 2023) | 0.2220 | 0.4501 | 0.0999 | 0.2802 | 0.1908 | 0.2578 | 0.0492 | 0.3795 | 0.1210 | 0.3244 | 0.0393 | 0.2532 |
| Qwen2-7B-Instruct (Yang et al., 2024) | 0.0000 | 0.2923 | 0.0000 | 0.0004 | 0.0002 | 0.4123 | 0.0001 | 0.0004 | 0.0000 | 0.0000 | 0.0000 | 0.0000 |
| Yi-1.5-9B (Young et al., 2024) | 0.2884 | 0.5461 | 0.1575 | 0.4927 | 0.2050 | 0.3724 | 0.0763 | 0.4126 | 0.1064 | 0.6596 | 0.0702 | 0.4526 |
| Chatglm-9B (GLM et al., 2024) | 0.3666 | 0.6902 | 0.2530 | 0.4736 | 0.3514 | 0.6820 | 0.2397 | 0.5000 | 0.1832 | 0.6506 | 0.1192 | 0.4342 |
| Llama3.2-1B-Instruct (Dubey et al., 2024) | 0.0644 | 0.5055 | 0.0326 | 0.1664 | 0.0822 | 0.4410 | 0.0363 | 0.1604 | 0.0714 | 0.4757 | 0.0340 | 0.1796 |
| Gemma3-12B (Team et al., 2025) | 0.4528 | 0.694 | 0.3142 | 0.5912 | 0.3256 | 0.3548 | 0.1155 | 0.4478 | 0.1666 | 0.0869 | 0.0145 | 0.3322 |
| Deepseek-R1-distill-Qwen-7B (Guo et al., 2025) | 0.4048 | 0.4416 | 0.1788 | 0.5622 | 0.2896 | 0.4535 | 0.1313 | 0.2034 | 0.0835 | 0.4556 | 0.0380 | 0.6074 |
| ChemDFM-v1.5-8B (Zhao et al., 2025) | 0.2998 | 0.6613 | 0.1983 | 0.7582 | 0.2510 | 0.6831 | 0.1715 | 0.8170 | 0.3112 | 0.6600 | 0.2054 | 0.8072 |
| ChemLLM-20B (Zhang et al., 2024) | 0.2992 | 0.5634 | 0.1686 | 0.3788 | 0.1316 | 0.5452 | 0.0717 | 0.293 | 0.2594 | 0.5739 | 0.1489 | 0.3856 |
| MolT5-small (Edwards et al., 2022) | 0.2158 | 0.1052 | 0.0227 | 0.4302 | 0.2316 | 0.1011 | 0.0234 | 0.4420 | 0.2214 | 0.1031 | 0.0228 | 0.4326 |
| MolT5-base (Edwards et al., 2022) | 0.2074 | 0.1051 | 0.0218 | 0.4168 | 0.1856 | 0.1073 | 0.0199 | 0.4796 | 0.2358 | 0.1054 | 0.0249 | 0.4536 |
| MolT5-large (Edwards et al., 2022) | 0.4244 | 0.1015 | 0.0431 | 0.8156 | 0.4496 | 0.1072 | 0.0482 | 0.8678 | 0.4654 | 0.1190 | 0.0554 | 0.9214 |
| BioT5-base (Pei et al., 2023) | 0.5158 | 0.1526 | 0.0787 | 1.0000 | 0.5060 | 0.1597 | 0.0808 | 1.0000 | 0.5068 | 0.1580 | 0.0801 | 1.0000 |
| Llama3.2-1B (OpenMolIns-large) | 0.2898 | 0.5951 | 0.1725 | 0.3850 | 0.2644 | 0.5956 | 0.1575 | 0.3678 | 0.1996 | 0.5849 | 0.1167 | 0.3490 |
| Llama3.1-8B (OpenMolIns-large) | 0.8054 | 0.6678 | 0.5378 | 0.8720 | 0.7122 | 0.6548 | 0.4663 | 0.8514 | 0.5224 | 0.6398 | 0.3342 | 0.8802 |
| Llama3.1-8B (OpenMolIns-xlarge) | 0.8822 | 0.6662 | 0.5877 | 0.9314 | 0.6982 | 0.6693 | 0.4673 | 0.9422 | 0.8648 | 0.6736 | 0.5825 | 0.9310 |
| Galactica-125M (OpenMolIns-light) | 0.3202 | 0.6547 | 0.2096 | 0.6416 | 0.3508 | 0.6435 | 0.2257 | 0.6358 | 0.2690 | 0.6521 | 0.1754 | 0.6380 |
| Galactica-125M (OpenMolIns-small) | 0.4172 | 0.6420 | 0.2678 | 0.5568 | 0.3958 | 0.6452 | 0.2554 | 0.5338 | 0.2956 | 0.6385 | 0.1887 | 0.5376 |
| Galactica-125M (OpenMolIns-medium) | 0.5904 | 0.5812 | 0.3431 | 0.7890 | 0.5874 | 0.5873 | 0.3450 | 0.7384 | 0.4608 | 0.5859 | 0.2700 | 0.7768 |
| Galactica-125M (OpenMolIns-large) | 0.6454 | 0.5927 | 0.3825 | 0.8198 | 0.6388 | 0.5973 | 0.3816 | 0.8028 | 0.4950 | 0.5962 | 0.2951 | 0.8100 |
| Galactica-125M (OpenMolIns-xlarge) | 0.7362 | 0.5744 | 0.4229 | 0.8902 | 0.7124 | 0.5697 | 0.4059 | 0.8612 | 0.5786 | 0.5677 | 0.3285 | 0.8626 |

Table 16: Detailed results on MolOpt. For each task, we highlight the best and the second-best success rate (SR), as well as the weighted success rate (WSR).

| Models | AtomNum | | | | BondNum | | | | FunctionalGroup | | | |
|---|---|---|---|---|---|---|---|---|---|---|---|---|
| | SR | Novelty | WSR | Validity | SR | Novelty | WSR | Validity | SR | Novelty | WSR | Validity |
| GPT-4o (OpenAI, 2023) | 0.1998 | 0.6703 | 0.1339 | 0.5852 | 0.0650 | 0.6336 | 0.0412 | 0.8564 | 0.2330 | 0.6513 | 0.1518 | 0.8590 |
| GPT-4-turbo (OpenAI, 2023) | 0.1702 | 0.6991 | 0.1190 | 0.4904 | 0.0774 | 0.6301 | 0.0488 | 0.9068 | 0.2180 | 0.6605 | 0.1440 | 0.8778 |
| GPT-3.5-turbo (OpenAI, 2023) | 0.1070 | 0.5054 | 0.0541 | 0.6947 | 0.0518 | 0.6871 | 0.0356 | 0.5522 | 0.1136 | 0.6585 | 0.0748 | 0.8686 |
| Claude-3.5 (Anthropic, 2024b) | 0.1928 | 0.6926 | 0.1335 | 0.6548 | 0.1058 | 0.6584 | 0.0697 | 0.8860 | 0.2364 | 0.6582 | 0.1556 | 0.8892 |
| Claude-3 (Anthropic, 2024a) | 0.1044 | 0.6833 | 0.0713 | 0.5910 | 0.1042 | 0.6598 | 0.0688 | 0.8696 | 0.1816 | 0.9158 | 0.1663 | 0.6644 |
| Gemini-1.5-pro (Deepmind, 2024) | 0.1808 | 0.6902 | 0.1202 | 0.6860 | 0.0708 | 0.6522 | 0.0462 | 0.8688 | 0.2486 | 0.6673 | 0.1659 | 0.9240 |
| Llama3-70B-Instruct (Int4) (Dubey et al., 2024) | 0.1404 | 0.6675 | 0.0937 | 0.5474 | 0.0670 | 0.6478 | 0.0434 | 0.7378 | 0.1752 | 0.6576 | 0.1152 | 0.7650 |
| Llama3-8B-Instruct (Dubey et al., 2024) | 0.0242 | 0.6649 | 0.0161 | 0.3812 | 0.0260 | 0.6303 | 0.0164 | 0.5700 | 0.0848 | 0.6167 | 0.0523 | 0.7216 |
| Llama3.1-8B-Instruct (Dubey et al., 2024) | 0.0228 | 0.7020 | 0.0160 | 0.3862 | 0.0395 | 0.6541 | 0.0258 | 0.6387 | 0.1300 | 0.6274 | 0.0816 | 0.6905 |
| Mistral-7B-Instruct-v0.2 (Jiang et al., 2023) | 0.0078 | 0.6732 | 0.0053 | 0.2986 | 0.0102 | 0.6309 | 0.0064 | 0.4524 | 0.0048 | 0.6012 | 0.0029 | 0.4020 |
| Qwen2-7B-Instruct (Yang et al., 2024) | 0.0110 | 0.9061 | 0.0100 | 0.2622 | 0.0010 | 0.8645 | 0.0090 | 0.0796 | 0.0022 | 0.8601 | 0.0019 | 0.0622 |
| Yi-1.5-9B (Young et al., 2024) | 0.0392 | 0.6848 | 0.0268 | 0.6170 | 0.0208 | 0.6407 | 0.0133 | 0.7072 | 0.0126 | 0.6945 | 0.0088 | 0.6521 |
| Chatglm-9B (GLM et al., 2024) | 0.0002 | 0.7483 | 0.0001 | 0.2131 | 0.0254 | 0.7189 | 0.0183 | 0.4682 | 0.0000 | 0.6908 | 0.0000 | 0.5926 |
| Llama3.2-1B-Instruct (Dubey et al., 2024) | 0.0040 | 0.6807 | 0.0027 | 0.1850 | 0.0080 | 0.7465 | 0.0060 | 0.2226 | 0.0008 | 0.7461 | 0.0006 | 0.2818 |
| Gemma3-12B (Team et al., 2025) | 0.0460 | 0.6017 | 0.0277 | 0.3414 | 0.0280 | 0.5973 | 0.0167 | 0.3318 | 0.0006 | 0.5940 | 0.0004 | 0.3224 |
| Deepseek-R1-distill-Qwen-7B (Guo et al., 2025) | 0.1574 | 0.6611 | 0.1041 | 0.4815 | 0.1631 | 0.7264 | 0.1185 | 0.5758 | 0.0385 | 0.6248 | 0.0241 | 0.5634 |
| ChemDFM-v1.5-8B (Zhao et al., 2025) | 0.0112 | 0.6809 | 0.0076 | 0.6700 | 0.0172 | 0.6146 | 0.0106 | 0.8968 | 0.0312 | 0.6011 | 0.0188 | 0.8094 |
| ChemLLM-20B (Zhang et al., 2024) | 0.0288 | 0.7055 | 0.0203 | 0.3530 | 0.0204 | 0.6786 | 0.0138 | 0.3678 | 0.0600 | 0.6452 | 0.0387 | 0.5442 |
| MolT5-small (Edwards et al., 2022) | 0.0006 | 0.6586 | 0.0004 | 0.6610 | 0.0064 | 0.5980 | 0.0038 | 0.6202 | 0.0114 | 0.5287 | 0.0060 | 0.8354 |
| MolT5-base (Edwards et al., 2022) | 0.0008 | 0.6868 | 0.0005 | 0.7560 | 0.0070 | 0.6509 | 0.0046 | 0.8422 | 0.0130 | 0.5464 | 0.0071 | 0.8382 |
| MolT5-large (Edwards et al., 2022) | 0.0150 | 0.7103 | 0.0107 | 0.9406 | 0.0118 | 0.5611 | 0.0066 | 0.8916 | 0.0382 | 0.6088 | 0.0233 | 0.9406 |
| BioT5-base (Pei et al., 2023) | 0.0118 | 0.8353 | 0.0099 | 0.9950 | 0.0078 | 0.6667 | 0.0052 | 0.9992 | 0.0476 | 0.6792 | 0.0323 | 0.9998 |
| Llama3.2-1B (OpenMolIns-large) | 0.0144 | 0.6490 | 0.0093 | 0.5616 | 0.0350 | 0.6150 | 0.0215 | 0.6186 | 0.0252 | 0.6373 | 0.0161 | 0.4412 |
| Llama3.1-8B (OpenMolIns-large) | 0.0136 | 0.6634 | 0.0090 | 0.7582 | 0.0544 | 0.6614 | 0.0360 | 0.7456 | 0.1344 | 0.6396 | 0.0860 | 0.6435 |
| Llama3.1-8B (OpenMolIns-xlarge) | 0.1186 | 0.6834 | 0.0811 | 0.8540 | 0.1196 | 0.6746 | 0.0807 | 0.9000 | 0.3548 | 0.6393 | 0.2268 | 0.9492 |
| Galactica-125M (OpenMolIns-light) | 0.0044 | 0.6054 | 0.0026 | 0.7930 | 0.0216 | 0.5724 | 0.0124 | 0.7596 | 0.0244 | 0.5756 | 0.0140 | 0.8442 |
| Galactica-125M (OpenMolIns-small) | 0.0146 | 0.6568 | 0.0096 | 0.8424 | 0.0530 | 0.6365 | 0.0337 | 0.7926 | 0.0570 | 0.5954 | 0.0339 | 0.8874 |
| Galactica-125M (OpenMolIns-medium) | 0.0294 | 0.6553 | 0.0193 | 0.8698 | 0.0622 | 0.6473 | 0.0403 | 0.7474 | 0.0882 | 0.6091 | 0.0537 | 0.8932 |
| Galactica-125M (OpenMolIns-large) | 0.0464 | 0.6729 | 0.0312 | 0.9116 | 0.0716 | 0.6695 | 0.0479 | 0.7374 | 0.0996 | 0.6276 | 0.0625 | 0.8966 |
| Galactica-125M (OpenMolIns-xlarge) | 0.1862 | 0.6899 | 0.1285 | 0.9308 | 0.1656 | 0.6887 | 0.1140 | 0.7952 | 0.2006 | 0.6445 | 0.1293 | 0.9162 |

Table 17: Detailed results for subtasks on MolCustom. For each task, we highlight the best and the second-best success rate (SR), as well as the weighted success rate (WSR).

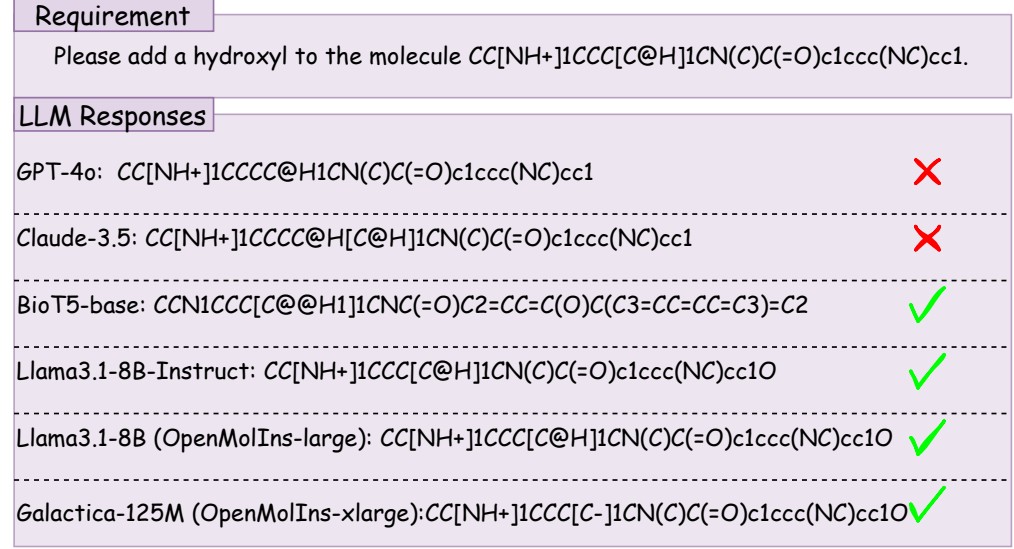

Figure 6: Case study of AddComponent. In this example, we require LLMs to add a hydroxyl to the given molecule. Here, GPT-4o and Claude-3.5 barely modified the molecule and failed to add a hydroxyl to the given molecule, while other four models successfully passed the testing process. However, it can be clearly noted that although BioT5-base passed the testing process, the generated molecule of BioT5-base could be quite different from the original molecule.

**Application to Complex Real-world Constraints.** Constraints like stereochemistry, 3D geometry, ring/bridge topology, and basic synthesizability represent important aspects of real-world molecular design. However, due to the complexity of these tasks, our work merely includes the counts of atoms, bonds, and functional groups to prob LLMs' understanding towards the molecular syntax, especially the details of the molecular structure. We believe that these complex constraints can be a promising future direction.

## H  BROADER IMPACTS

$S^2$-Bench firstly proposes the text-based open molecule generation task for LLMs, aiming at adopting LLMs as chemist assistants and molecule operators for molecule discovery. Hopefully, our benchmark could inspire more researches in this promising area, thereby accelerating the pace of drug discovery and material design. Ultimately, these development in molecule discovery will benefit the human welfare in various ways. For example, the effective medicines could help save people's lives and expand the lifespan.

## I  USE OF LLMS

During the preparation of this work, the author(s) used LLMs to improve the language and readability. After using this tool/service, the author(s) reviewed and edited the content as needed and take(s) full responsibility for the content of the publication.

## J  CASE STUDY

Finally, we include the case study to better illustrate the performance gap among different models.

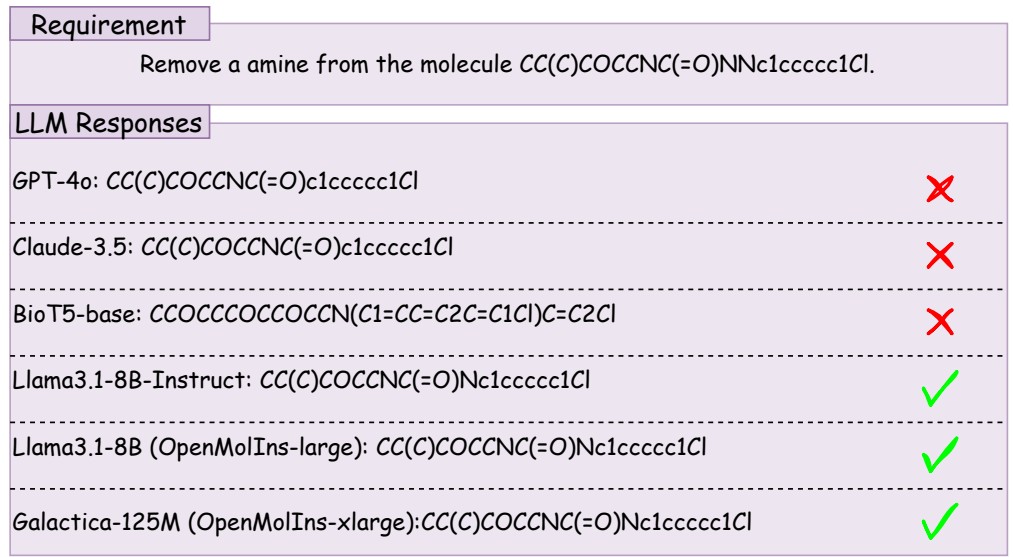

Figure 7: Case study of DelComponent. In this example, we require LLMs to remove an amine from the given molecule. Here, GPT-4o, Claude-3.5, and BioT5-base all removed two amines, while the remaining models successfully passed the testing process.

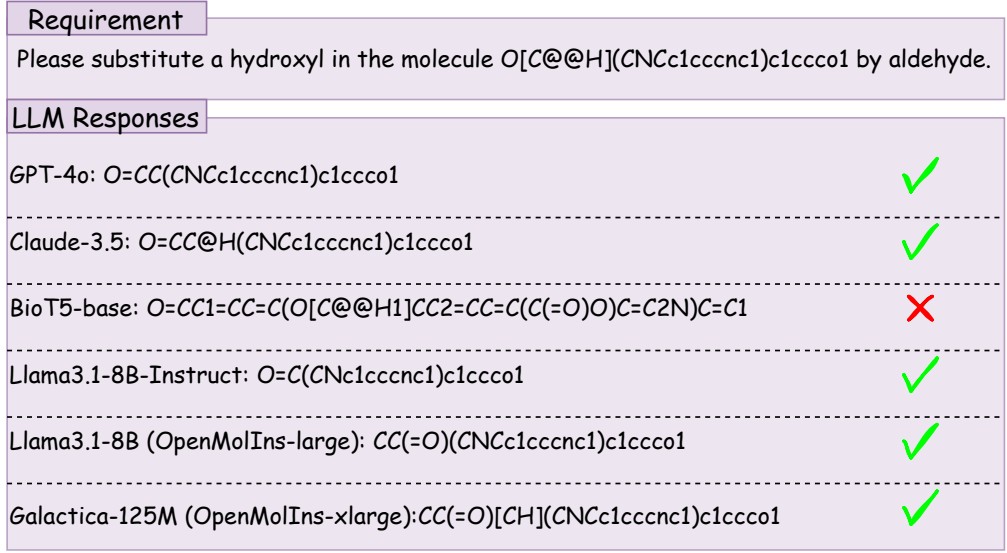

Figure 8: Case study of SubComponent. In this example, we require LLMs to substitute the hydroxyl from the given molecule by a aldehyde. Here, although BioT5-base added a aldehyde, it failed to remove the hydroxyl. In contrast, the remaining models successfully passed the testing process.

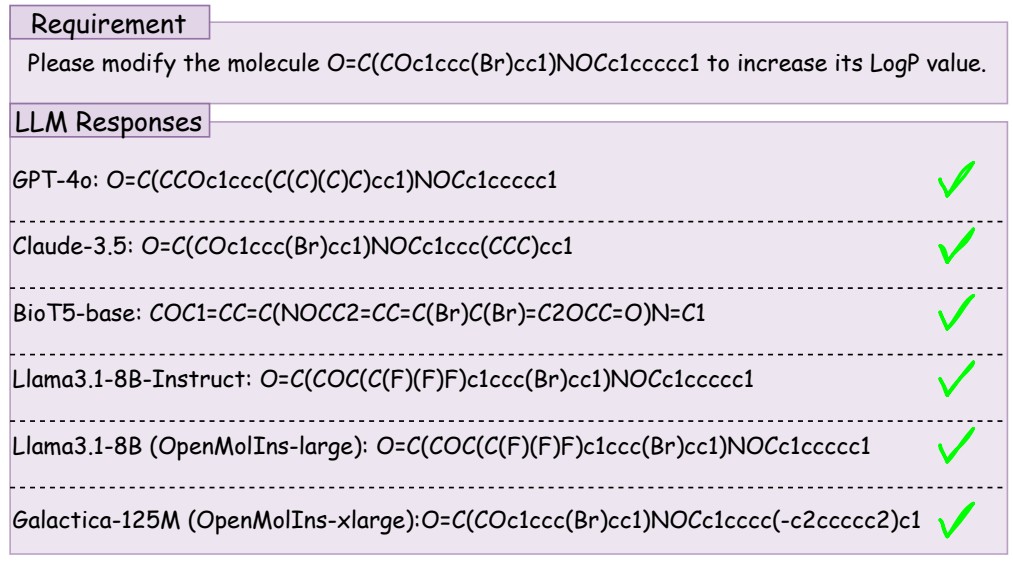

Figure 9: Case study of LogP. In this example, we require LLMs to modfify the molecule to have a higher LogP value. Here, after the calculation of LogP values, we found all the generated molecules satisfy the requirement.

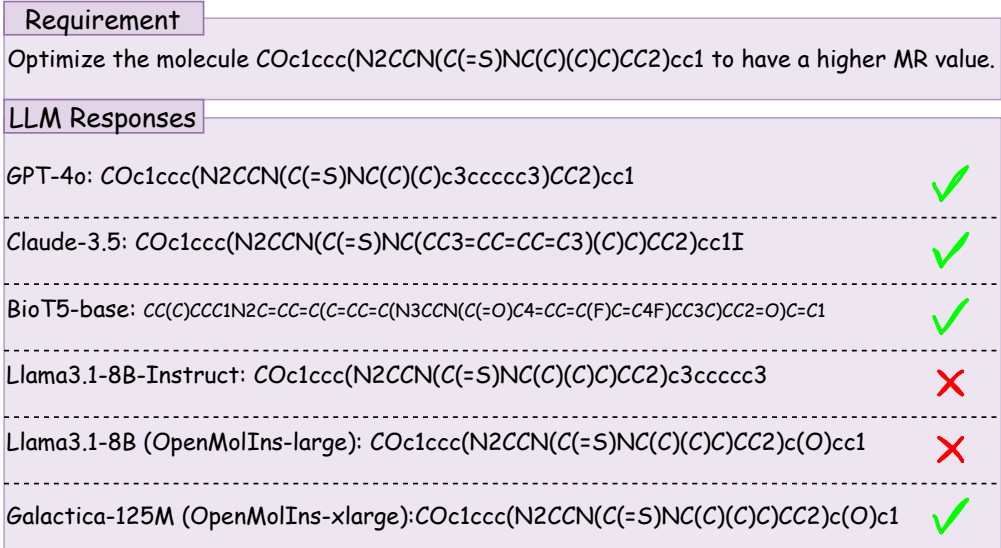

Figure 10: Case study of MR. In this example, we require LLMs to modify the molecule to have a higher MR value. Here, Llama3.1-8B-Instruct and Llama3.1-8B-Instruct (OpenMolIns-large) surprisingly failed the testing process. From the two errors, we could observe that the two models were trying to introduce extra rings to increase the MR value, while the ring numbers were misassigned.

**Requirement**

Optimize the molecule C#CCNCC(=O)N[C@H](C)Cc1ccco1 to have a lower QED value.

**LLM Responses**

GPT-4o: C#CCNCC(=O)NC@HCc1ccc(c2ccccc2)O     ✗

----

Claude-3.5: C#CCNCC(=O)NC(CCl)(c1c(Br)cc(I)cc1Cl)c2ccccc2     ✓

----

BioT5-base: CC#CC=C(CCCNC(=O)C(C)CC1)CC=CO1     ✗

----

Llama3.1-8B-Instruct: CCNCC(NC(C)C1=CC=CO1)C(=O)N     ✗

----

Llama3.1-8B (OpenMolIns-large): C#CCNCC(=O)NC=C(C)Cc1ccco1     ✗

----

Galactica-125M (OpenMolIns-xlarge): C#CCNCC(=O)N[AsH](C)Cc1cc(O)co1     ✓

Figure 11: Case study of QED. In this example, we require LLMs to lower the QED value of the given molecule. Compared to LogP and MR, QED is more difficult to predict. We could observe that GPT-4o, BioT5-base, Llama3.1-8B-Instruct, and Llama3.1-8B-Instruct (OpenMolIns-large) all increased the QED value of the molecule, while Claude-3.5 and Galactica-125M (OpenMolIns-xlarge) successfully captured the requirement and lowered the QED value.

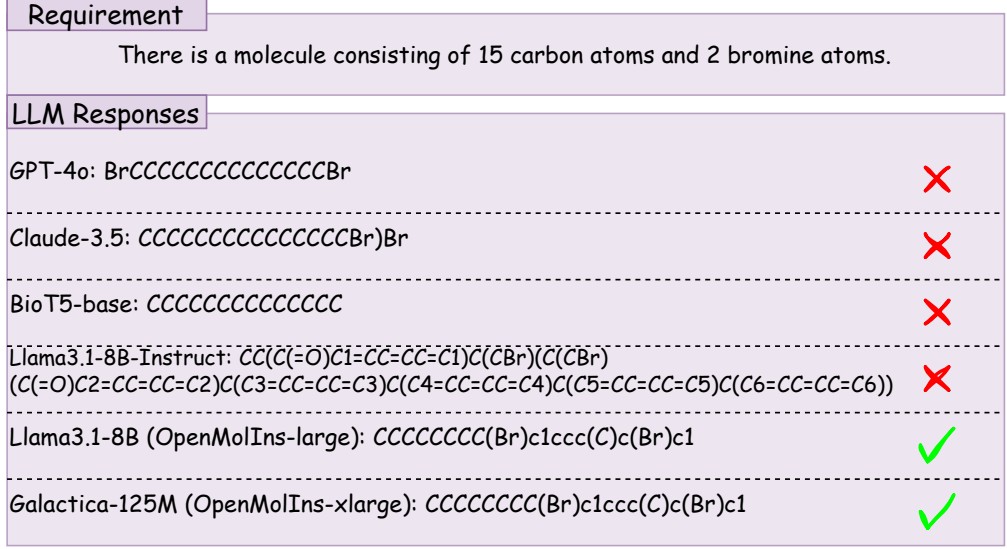

**Requirement**

There is a molecule consisting of 15 carbon atoms and 2 bromine atoms.

**LLM Responses**

GPT-4o: BrCCCCCCCCCCCCCCBr     ✗

----

Claude-3.5: CCCCCCCCCCCCCCCBr)Br     ✗

----

BioT5-base: CCCCCCCCCCCCCCC     ✗

----

Llama3.1-8B-Instruct: CC(C(=O)C1=CC=CC=C1)C(CBr)(C(CBr) (C(=O)C2=CC=CC=C2)C(C3=CC=CC=C3)C(C4=CC=CC=C4)C(C5=CC=CC=C5)C(C6=CC=CC=C6))     ✗

----

Llama3.1-8B (OpenMolIns-large): CCCCCCCC(Br)c1ccc(C)c(Br)c1     ✓

----

Galactica-125M (OpenMolIns-xlarge): CCCCCCCC(Br)c1ccc(C)c(Br)c1     ✓

Figure 12: Case study of AtomNum. In this example, we require LLMs to generate a molecule with 15 carbon atoms and 2 bromine atoms. Here, GPT-4o, Claude-3.5, BioT5-base, and Llama3.1-8B-Instruct all generated incorrect numbers of the specified atoms.

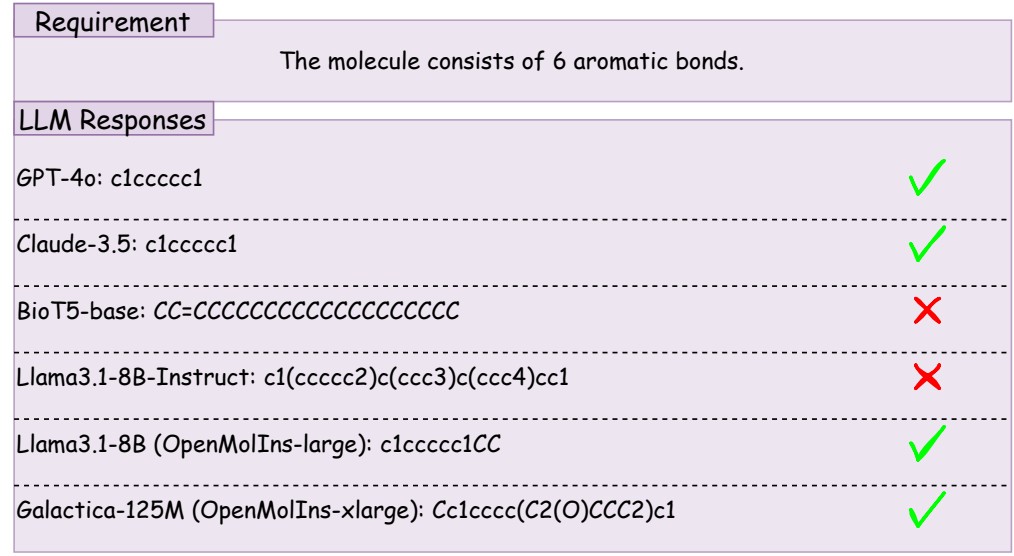

Figure 13: Case study of BondNum. In this example, we require LLMs to generate a molecule with 6 aromatic bonds, which normally exist in benzene rings. Here, GPT-4o, Claude-3.5, Llama3.1-8B-Instruct (OpenMolIns-large), and Galactica-125M (OpenMolIns-xlarge) captured the concept and generated molecules with a benzene ring in their structures. In contrast, BioT5-base failed to identify the requirement and Llama3.1-8B-Instruct generated too may aromatic rings.

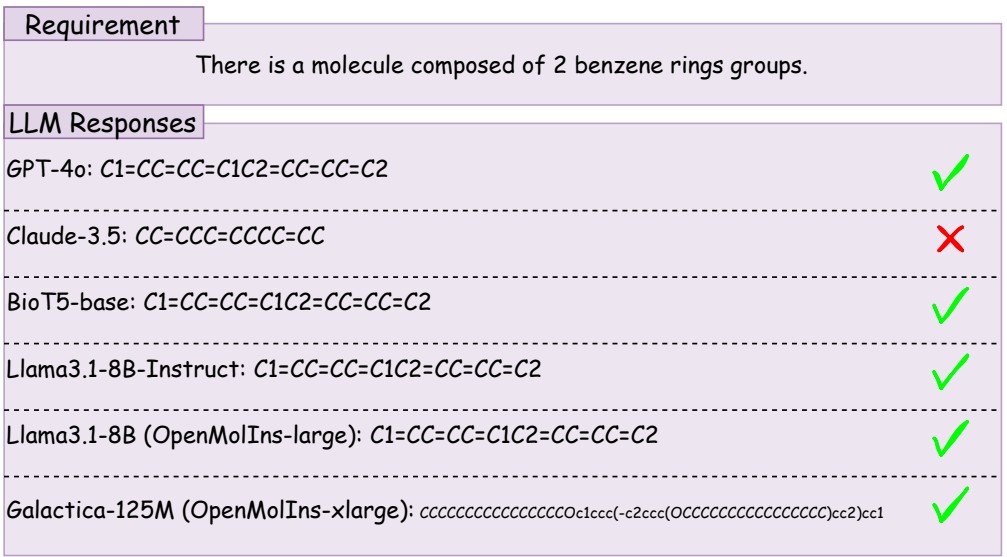

Figure 14: Case study of FunctionalGroup. In this example, we require LLMs to generate a molecule with 2 benzene rings. Here, only Claude-3.5 failed to generate the correct number of benzene rings.

