# OpenReview forum: "Speak-to-Structure: Evaluating LLMs in Open-domain Natural Language-Driven Molecule Generation"
_ICLR.cc/2026/Conference — Submitted to ICLR 2026_

### Official Review · Reviewer_Cos8 · 2025-10-28

**Soundness:** 1
**Presentation:** 2
**Contribution:** 1
**Rating:** 2
**Confidence:** 4

**Summary:**

The author suggests that LLM-based molecular generation should adhere to a one-to-many principle and has constructed a benchmark dataset based on this principle. However, the core concepts and dataset construction method of this work are very similar to those in article [1]. However, the manuscript does not appropriately cite [1] or articulate the differences between the two works. Additionally, the LLMs evaluated in this work are all general-purpose LLMs. While they possess some understanding of SMILES data, there remains a significant gap with the molecular generation field. The author should assess more domain-specific LLMs.

[1] https://link.springer.com/article/10.1186/s12915-025-02200-3

**Strengths:**

The principle that molecular generation should adhere to a one-to-many approach is crucial, and the author has developed a benchmark based on this notion, evaluating it across multiple LLMs. However, the core concept and the method of constructing the benchmark in this paper are highly similar to those in work [1].

**Weaknesses:**

The methodology is highly similar to that in work [1] and lacks comprehensive research. The LLMs evaluated are all general-purpose models, with a noticeable absence of specialized domain-specific models.

**Questions:**

no

---

> ### Author Response · Authors · 2025-11-17
>
> >W1: The methodology is highly similar to that in work [1] and lacks comprehensive research.
>
> #### Our Response:
>
> Thank you for letting us know the potential similarity with the cited work [1], which is an excellent work focused on multi-constraint molecular generation problem. We apologize if our manuscript did not sufficiently highlight the distinctions, and we are grateful for the opportunity to clarify.
>
> First of all, we would like to state for the record that our work was conducted independently and completed prior to the publication of the paper you mentioned. We were not aware of this work during our research and development process. Nevertheless, we are, of course, happy to include a citation to this excellent work and add a comparative discussion in our revised manuscript.
>
> However, we wish to respectfully emphasize that our work is fundamentally different in several key aspects, including our core motivation, task scope, dataset formation, and primary contributions. We believe these differences are significant:
> 1. **Core Motivation**: Our work is focused on **open-ended** nature of molecular generation, while the cited work focuses on **multi-constrained** generation. Our goal is to examine LLMs' genuine molecular understanding and generation capabilities. In contrast, the cited work aims to propose a model that precisely satisfies a specific set of predefined properties and structural constraints simultaneously.
> 2. **Task Scope**: Our task requires LLMs to develop a **deep and granular understanding of molecular structure and syntax**, including *atoms*, *bonds*, and *functional groups* (MolCustom), to perform **precise edits** in the molecular strcuture (MolEdit), and to **optimize molecular properties** via structural edits (MolCustom). In contrast, the cited work’s scope is primarily focused on the level of **functional groups and direct estimation of molecular properties**.
> 3. **Dataset Formation**: Our approach constructs datasets using a **diverse pool of templates** for each subtask. Especially for MolCustom, we **infill the templates with random numbers from predefined distributions**. In contrast, the cited work employs a teacher-student framework. In this framework, **teachers first extract molecular knowledge** (e.g., structure, properties, binding affinity) to form **multi-constraint requirements using task-specific templates** (one template per task). While both works use templates to generate instructions, they are considered as a standard practice for creating language-based tasks [2,3,4]. Meanwhile, a key difference lies in the input of our MolEdit and MolOpt tasks that **the original molecule is also infilled as the input**, whereas the cited work relies **solely on natural language descriptions**.
> 5. **Primary Contributions**: Our main contribution lies in a **novel benchmark/dataset**, a systematic **automatic evaluation process** for text-based **open-domain molecular generation**, providing **insights regarding the chemical capabilities of existing LLMs**. In contrast, the contribution of the cited work lies in proposing a **new teacher-student model architecture**, constructing a **large-scale text-molecule dataset**, and validating a model’s capability on **multi-constraint generation tasks**.
>
> We sincerely hope this clarification addresses your concerns and we have updated the manuscript to showcase these distinctions between the two works. The differences in motivation, task scope, dataset formation, and contribution are substantial. We believe our work offers a unique and complementary perspective to the bio-molecular field. Based on these points, we kindly request that you reconsider our work in light of its distinct contributions.

---

> ### Author Response · Authors · 2025-11-17
>
> >W2: The LLMs evaluated are all general-purpose models, with a noticeable absence of specialized domain-specific models.
>
> #### Our Response:
> Thank you again for your constructive suggestion. We fully agree with you that specialized domain-specific models are essential for evaluation.
>
> In this benchmark, we did include several specialized domain-specific models like MolT5 [5] and BioT5 [6] (grouped as Open-source ChEBI-20 Fine-tuned LLMs). However, we do notice that there are recently several chemical LLMs like ChemLLM [7] and ChemDFM[8], and we appreciate your suggestion to include them in our benchmark. These models are classified as Open-source General & Chemical LLMs. Specifically, their detailed performance are listed in the table below.
>
> | MolEdit         | AddComponent |            |            |          | DelComponent |            |            |          | SubComponent    |            |            |          |
> |-----------------|--------------|------------|------------|----------|--------------|------------|------------|----------|-----------------|------------|------------|----------|
> | Models          | SR           | Similarity | WSR        | Validity | SR           | Similarity | WSR        | Validity | SR              | Similarity | WSR        | Validity |
> | ChemLLM-20B     | 0.1436       | 0.5978     | 0.0858 | 0.2108   | 0.2092       | 0.5734     | 0.1200 | 0.3098   | 0.3088          | 0.6805     | 0.2101  | 0.4936   |
> | ChemDFM-v1.5-8B | 0.2942         | 0.6454       | 0.1899     | 0.9112     | 0.2056         | 0.6937       | 0.1426     | 0.9286     | 0.2198            | 0.6432       | 0.1414     | 0.8652     |
>
> | MolOpt          | LogP         |            |            |          | MR           |            |            |          | QED             |            |            |          |
> |-----------------|--------------|------------|------------|----------|--------------|------------|------------|----------|-----------------|------------|------------|----------|
> | Models          | SR           | Similarity | WSR        | Validity | SR           | Similarity | WSR        | Validity | SR              | Similarity | WSR        | Validity |
> | ChemLLM-20B     | 0.2992       | 0.5634     | 0.1686 | 0.3788   | 0.1316       | 0.5452     | 0.0717 | 0.293    | 0.2594          | 0.5739     | 0.1489 | 0.3856   |
> | ChemDFM-v1.5-8B | 0.2998          | 0.6613       | 0.1983      | 0.7582     | 0.2510         | 0.6831      | 0.1715       | 0.8170     | 0.3112            | 0.6600       | 0.2054     | 0.8072     |
>
> | MolCustom       | AtomNum      |            |            |          | BondNum      |            |            |          | FunctionalGroup |            |            |          |
> |-----------------|--------------|------------|------------|----------|--------------|------------|------------|----------|-----------------|------------|------------|----------|
> | Models          | SR           | Novelty    | WSR        | Validity | SR           | Novelty    | WSR        | Validity | SR              | Novelty    | WSR        | Validity |
> | ChemLLM-20B     | 0.0288       | 0.7055     | 0.0203  | 0.3530    | 0.0204       | 0.6786     | 0.0138 | 0.3678   | 0.0600            | 0.6452     | 0.0387   | 0.5442   |
> | ChemDFM-v1.5-8B | 0.0112         | 0.6809       | 0.0076     | 0.6746     | 0.0172         | 0.6146      | 0.0106     | 0.8968      | 0.0312            | 0.6011        | 0.0188      | 0.8094     |
>
> Correspondingly, the two chemical LLMs obtains 21-st and 22-nd place in our benchmark, which showcases the potential limitation of the previous chemical pre-training. We will update the manuscript to clearly demonstrate the performance of these models.

---

> ### Author Response · Authors · 2025-11-17
>
> #### References
> [1] Zhou, P., Wang, J., Li, C., Wang, Z., Liu, Y., Sun, S., ... & Zeng, X. (2025). Instruction multi-constraint molecular generation using a teacher-student large language model. BMC biology, 23(1), 105.
>
> [2] Cui, L., Wu, Y., Liu, J., Yang, S., & Zhang, Y. (2021, August). Template-Based Named Entity Recognition Using BART. In Findings of the Association for Computational Linguistics: ACL-IJCNLP 2021 (pp. 1835-1845).
>
> [3] Lu, P., Mishra, S., Xia, T., Qiu, L., Chang, K. W., Zhu, S. C., ... & Kalyan, A. (2022). Learn to explain: Multimodal reasoning via thought chains for science question answering. Advances in Neural Information Processing Systems, 35, 2507-2521.
>
> [4] Fang, Y., Liang, X., Zhang, N., Liu, K., Huang, R., Chen, Z., ... & Chen, H. Mol-Instructions: A Large-Scale Biomolecular Instruction Dataset for Large Language Models. In The Twelfth International Conference on Learning Representations.
>
> [5] Edwards, C., Lai, T., Ros, K., Honke, G., Cho, K., & Ji, H. (2022, December). Translation between Molecules and Natural Language. In Proceedings of the 2022 Conference on Empirical Methods in Natural Language Processing (pp. 375-413).
>
> [6] Pei, Q., Zhang, W., Zhu, J., Wu, K., Gao, K., Wu, L., ... & Yan, R. (2023, December). BioT5: Enriching Cross-modal Integration in Biology with Chemical Knowledge and Natural Language Associations. In Proceedings of the 2023 Conference on Empirical Methods in Natural Language Processing (pp. 1102-1123).
>
> [7] Zhang, D., Liu, W., Tan, Q., Chen, J., Yan, H., Yan, Y., ... & Li, Y. (2024). Chemllm: A chemical large language model. arXiv preprint arXiv:2402.06852.
>
> [8] Zhao, Z., Ma, D., Chen, L., Sun, L., Li, Z., Xia, Y., ... & Chen, X. (2025). Developing ChemDFM as a large language foundation model for chemistry. Cell Reports Physical Science, 6(4).

---

> > ### Comment · Reviewer_Cos8 · 2025-11-27
> >
> > Thank you to the author for the response. Firstly, I noticed that the preprint of work [1] was released on March 20, 2024, with the official version published on April 23, 2025. Secondly, while putting aside the similarities in methodology, your work encompasses not only molecular generation but also molecular editing and optimization based on large language models.  However, there are significant pieces of research on these two tasks that have not been mentioned, such as [2][3]. Thirdly, I have not seen the revised version yet. I am aware that this project has required substantial time and effort, but conducting a thorough review of existing work is a fundamental responsibility in conducting responsible research.
> >
> > [2] Wu, Zhenxing, et al. "Leveraging language model for advanced multiproperty molecular optimization via prompt engineering." Nature Machine Intelligence 6.11 (2024): 1359-1369.
> > [3] Ye, Geyan, et al. "Drugassist: A large language model for molecule optimization." Briefings in Bioinformatics 26.1 (2025): bbae693.

---

> ### Author Response · Authors · 2025-11-28
>
> Thank you so much for the following response! We are committed to resolve your remaining concerns.
>
> First of all, we mention the timeline just for clarification and we are indeed not aware of this excellent work [1] before your comments. We sincerely hope to clarify any misunderstandings:
> * We clarify that the similarity originates from **the nature of text-guided molecular generation** and **the idea of applying LLMs as chemist assistants**.
> * Our work is focused on the **open generation** of molecules, instead of **multi-constraint molecular optimization**.
> * We provide a new molecular **dataset and benchmark**, aiming at a novel aspect for testing the capability of LLMs in the biomolecular field and inspiring more works in this field.
>
> That is to say, there is **no fundamental conflict** between the novel model/methodology proposed in [1] and our benchmark work.
>
> Regarding the lack of reviewing existing works [2] [3], we are sorry for our ignorance and really appreciate your suggestions. We find that our lack of reviewing these works is mainly due to our task focus (text-based molecule generation [4] and multi-objective molecular optimization [5]). We believe your comment provides a new aspect for us to comprehensively review the related works. We have then conducted a more comprehensive review of these works, highlighting the differences between them and our work.
>
> We have revised our manuscript now for your reference. Additionally, we find the works [6] [7] are also of great importance and should be mentioned and discussed carefully in our work.
>
> Nevertheless, we truly apologize that our work did not mention nor discussed the work [1] due to our ignorance. We sincerely hope that our explanations and revisions could resolve your concerns.
>
> #### References
> [1] Zhou, P., Wang, J., Li, C., Wang, Z., Liu, Y., Sun, S., ... & Zeng, X. (2025). Instruction multi-constraint molecular generation using a teacher-student large language model. BMC biology, 23(1), 105.
>
> [2] Wu, Z., Zhang, O., Wang, X., Fu, L., Zhao, H., Wang, J., ... & Hou, T. (2024). Leveraging language model for advanced multiproperty molecular optimization via prompt engineering. Nature Machine Intelligence, 6(11), 1359-1369.
>
> [3] Ye, G., Cai, X., Lai, H., Wang, X., Huang, J., Wang, L., ... & Zeng, X. (2025). Drugassist: A large language model for molecule optimization. Briefings in Bioinformatics, 26(1), bbae693.
>
> [4] Edwards, C., Lai, T., Ros, K., Honke, G., Cho, K., & Ji, H. (2022, December). Translation between Molecules and Natural Language. In Proceedings of the 2022 Conference on Empirical Methods in Natural Language Processing (pp. 375-413).
>
> [5] Nicolaou, C. A., & Brown, N. (2013). Multi-objective optimization methods in drug design. Drug Discovery Today: Technologies, 10(3), e427-e435.
>
> [6] Zhou, P., Tim, L. H., Cheng, Z., Xie, K., Li, C., Liu, W., & Zeng, X. (2025). Enhancing Molecular Property Prediction with Knowledge from Large Language Models. arXiv preprint arXiv:2509.20664.
>
> [7] Zhuang, X., Ding, K., Lyu, T., Jiang, Y., Li, X., Xiang, Z., ... & Chen, H. (2025). Advancing biomolecular understanding and design following human instructions. Nature Machine Intelligence, 7(7), 1154-1167.

---

### Official Review · Reviewer_fcDQ · 2025-10-30

**Soundness:** 4
**Presentation:** 3
**Contribution:** 3
**Rating:** 6
**Confidence:** 4

**Summary:**

This paper introduces S²-Bench (Speak-to-Structure), a large-scale benchmark designed to evaluate the capability of LLMs in open-domain natural-language-driven molecular generation. Unlike traditional one-to-one text-to-SMILES datasets, S²-Bench adopts a one-to-many paradigm, allowing multiple valid molecular outputs for a single instruction to better mimic real-world chemical design. The benchmark includes three complementary tasks—MolEdit, MolOpt, and MolCustom—that respectively assess structural editing, property optimization, and constrained generation. To support this evaluation, the authors construct OpenMolIns, a 1.2-million-sample programmatically generated dataset of instruction–molecule pairs using RDKit-based property computation and LLM paraphrasing for linguistic diversity. The paper further proposes the Weighted Success Rate (WSR) metric to integrate chemical validity, success rate, and novelty. Experiments on 28 LLMs (e.g., GPT-4o, Claude-3.5, LLaMA-3.1, Qwen-2) reveal that current models struggle with true chemical reasoning, while instruction-tuned variants significantly outperform base models. Overall, the work provides a reproducible, scalable, and realistic benchmark for assessing how LLMs perform in molecular reasoning and creative design.

**Strengths:**

The paper makes a timely and impactful contribution by redefining how natural-language-driven molecule generation should be evaluated.
(1) The benchmark is conceptually original, introducing the one-to-many mapping paradigm that better reflects chemical diversity and real-world design scenarios.
(2) The dataset generation pipeline is well-engineered and reproducible, integrating chemical computation (RDKit) with LLM-based linguistic diversification.
(3) The evaluation design—combining MolEdit, MolOpt, and MolCustom—offers a comprehensive assessment of LLMs’ structural reasoning and control capabilities.
(4) Experiments are large-scale and convincing, covering both open and closed LLMs with consistent results.
(5) The paper is clearly written, well-structured, and likely to serve as a standard benchmark for future research in chemical LLMs.

**Weaknesses:**

The main weakness lies in the limited methodological depth. While the benchmark is well-designed, the work does not provide theoretical insight into the relationship between language semantics and chemical structure reasoning. The Weighted Success Rate metric, though practical, appears heuristic, and its weighting choices are not empirically justified. The programmatic data generation may also introduce semantic drift between the instruction and molecule, especially after paraphrasing by LLMs. Additionally, the dataset relies exclusively on RDKit-computed properties, which may not reflect experimental conditions, and the benchmark focuses primarily on organic drug-like molecules, limiting generalization to other domains.

**Questions:**

Q1. Data reliability after paraphrasing
Since all instructions are automatically generated and paraphrased by LLMs, how do the authors ensure that the final language remains semantically consistent with the intended molecular transformation? Would human verification or semantic-similarity filtering improve dataset fidelity?

Q2. Distributional bias of molecular sources
OpenMolIns is built mainly from ZINC, ChEMBL, and MOSES, which are biased toward drug-like molecules. Could this limit generalization to other chemical domains (e.g., materials or catalysts)?

Q3. Scientific validity of the evaluation metric
How are the weights in the Weighted Success Rate (WSR) determined, and do they align with human expert judgments of chemical success or usefulness?

Q4. Language understanding vs. template learning
Since instructions are generated from deterministic templates, does S²-Bench truly measure language understanding or just template matching? Have the authors tested models on non-templated, free-form instructions?

Q5. Out-of-distribution generalization
Have the authors evaluated the benchmark under compositional or OOD instructions, e.g., “Add a hydroxyl group while keeping molecular weight below 200”?

Q6. Alignment with real chemical measurements
Since all target properties come from RDKit computations, how well do they correlate with experimental measurements? Could integrating experimental datasets improve realism?

---

> ### Author Response · Authors · 2025-11-21
>
> We sincerely thank you for your constructive and insightful feedback. We are particularly encouraged by your recognition of S²-Bench’s significance, real-world relevance, dataset creation pipeline, automated evaluation process, and comprehensive experimental findings. We are glad to have the opportunity to clarify your concerns.
>
> >W1: The main weakness lies in the limited methodological depth. While the benchmark is well-designed, the work does not provide theoretical insight into the relationship between language semantics and chemical structure reasoning.
>
> #### Our Response:
> Thank you for highlighting the methodological aspect. We agree that our current work focuses primarily on benchmarking rather than providing deep theoretical analysis.
>
> We would like to clarify that the primary methodological contribution of our work is the design of the S²-Bench and its evaluation framework:
> * The task design of our benchmark is deliberate. From MolEdit, MolOpt to MolCustom, the task difficulty gradually improves and each task mirrors a critical phase of real-world molecule discovery.
>     * **MolEdit** assesses the foundational capability for "precise, localized structural modifications" , such as adding, deleting, or substituting molecular components.
>     * **MolOpt** extends the MolEdit task and takes a step further. It requires LLMs to perform "goal-oriented chemical reasoning", challenging them to not only edit a structure but also to ensure the modification "leads to a desired change in a specific property".
>     * **MolCustom** serves as the "ultimate test of an LLM's creative chemical design ability". It requires the model to synthesize a novel molecule from scratch based on a set of "quantitative and qualitative constraints" , such as a defined number of atoms (AtomNum), bonds (BondNum), or functional groups (FunctionalGroup).
> * Meanwhile, we move beyond the targeted molecular generation of previous datasets, which we argue inadvertently encourage memorization and still utilizes traditional translation metrics for evaluation. Our benchmark, introducing the open-domain molecular generation, specifically engineered to shift the evaluation from "simple pattern recall" to "realistic molecular design".
>
> That said, we acknowledge that the benchmark could be further expanded to include more complex and diverse structural categories (e.g., specific ring systems, polymers, or large macrocycles) to more deeply probe an LLM's understanding and generation capabilities for these challenging structures.
>
> Finally, thank you again for your insightful comments and valuable feedback on our work.

---

> > ### Author Response · Authors · 2025-11-21
> >
> > >W4 & Q6: Additionally, the dataset relies exclusively on RDKit-computed properties, which may not reflect experimental conditions,
> > >Alignment with real chemical measurements
> > Since all target properties come from RDKit computations, how well do they correlate with experimental measurements? Could integrating experimental datasets improve realism?
> >
> > #### Our Response:
> > We appreciate your constructive comments regarding the reliance on RDKit-computed properties in our benchmark. We fully agree with you that more diverse properties will significantly benefit the width of our benchmark.
> >
> > However, the discrepancy between computationally derived descriptors and experimentally measured properties remains a fundamental open question in computational chemistry, with no broadly accepted solution at present. Within this context, we made our best effort to select properties that are meaningful while ensuring scientific rigor, reproducibility, and scalability.
> >
> > In our benchmark, we chose RDKit-computed properties such as LogP, QED, and MR because:
> > 1. They are deterministic and fully reproducible. Any researcher can reproduce our exact evaluation results using the same open-source tool, ensuring fairness.
> > 2. They are computationally inexpensive. This allows us to evaluate thousands of generated molecules, which is infeasible with expensive experimental or high-level quantum mechanical calculations.
> > 3. They are widely accepted properties. While not perfect, properties like LogP and QED are standard, well-understood metrics in the field that serve as a consistent baseline for evaluating a model’s chemical reasoning capabilities.
> >
> > We also explored the inclusion of more complex or experimentally closer properties (e.g., toxicity-related or bioactivity-related descriptors) during the design phase. Ultimately, we decided not to formally include them because their computational accuracy is difficult to justify, and they do not provide sufficiently representative or consistent evaluations across chemical space.
> >
> > Nevertheless, we agree that incorporating such properties as reference can be valuable. Therefore, we will release these additional properties as optional reference, allowing researchers to explore alternative evaluation perspectives without compromising the benchmark’s reproducibility.

---

> ### Author Response · Authors · 2025-11-21
>
> >W2 & Q3: The Weighted Success Rate metric, though practical, appears heuristic, and its weighting choices are not empirically justified.
> >Scientific validity of the evaluation metric
> How are the weights in the Weighted Success Rate (WSR) determined, and do they align with human expert judgments of chemical success or usefulness?
>
>
> #### Our Response:
>
> Thank you for your insightful comment. In fact, incorporating weights is essential for ensuring meaningful success rather than merely passing the automatic testing process. In MolEdit and MolOpt tasks, the models are allowed to make extra modifications to ensure the chemical validity, so the exact edit path on SMILES strings cannot be traced. Therefore, the similarity helps verify that the model’s generation is truly derived from the input molecule rather than an unrelated structure.
>
> For instance, we can take a further look at the performance of BioT5-base in MolEdit and MolOpt. For comparison, we randomly sampled 100 examples each subtask for expert to evaluate the correctness and quality of the generation. Specifically, human experts give a score ranging from 0 to 5 for each example. Here,
> * 5 denotes a perfect success,
> * 4 means success but with redundant operations (a little different structure),
> * 3 shows failure but modified from the original molecule (similar structure),
> * 2 denotes failure and a little different structure,
> * 1 indicates success but unrelated/different structure,
> * 0 denotes failure and unrelated/different structure.
>
> We caculate the average score of expert evaluation.
>
> | BioT5-base |    Subtasks          | SR     | Similarity | WSR    | Validity | Expert Evaluation |
> |------------|--------------|--------|------------|--------|----------|----------|
> | MolEdit    | AddComponent | 0.3462 | 0.1567     | 0.0542 | 1        | 0.64        |
> |            | DelComponent | 0.1668 | 0.1597     | 0.0266 | 1        | 0.22        |
> |            | SubComponent | 0.0684 | 0.1576     | 0.0108 | 0.9998   | 0.19        |
> | MolOpt     | LogP         | 0.5158 | 0.1526     | 0.0787 | 1        | 0.58        |
> |            | MR           | 0.506  | 0.1597     | 0.0808 | 1        | 0.67        |
> |            | QED          | 0.5068 | 0.158      | 0.0801 | 1        | 0.64        |
>
> We can see that although BioT5-base achieves high success rates for MolEdit and MolOpt, the similarity scores are extremely low, indicating that the generated molecules are not modified from the original molecule but from scratch.
>
> To assess which metric better reflects expert judgment, we computed both Pearson and Spearman correlations between SR/WSR and the expert evaluation scores across the subtasks. The results show that WSR exhibits consistently stronger alignment with expert ratings. Specifically, WSR achieves a slightly higher Pearson correlation with expert scores (r = 0.926 vs. 0.922 for SR) and, more importantly, a substantially higher Spearman rank correlation (ρ = 0.899 vs. 0.551 for SR). While the correlation between SR and expert ratings is not statistically significant (p = 0.257), WSR shows strong and significant rank consistency (p = 0.015).
>
> At the same time, for the MolCustom task, novelty is widely used for evaluating the quality of generated molecules [1].
>
> Therefore, it is essential to have these weights to reflect the real capabilities of LLMs more faithfully.

---

> ### Author Response · Authors · 2025-11-21
>
> >W3 & Q1: The programmatic data generation may also introduce semantic drift between the instruction and molecule, especially after paraphrasing by LLMs.
> >Data reliability after paraphrasing
> Since all instructions are automatically generated and paraphrased by LLMs, how do the authors ensure that the final language remains semantically consistent with the intended molecular transformation? Would human verification or semantic-similarity filtering improve dataset fidelity?
>
> #### Our Response:
> Thank you for raising this important point regarding potential semantic drift. We agree that this is a critical concern in programmatic data generation. However, we would like to take this opportunity to clarify a key aspect of our methodology to avoid any misunderstanding.
>
> Our pipeline does not involve using LLMs to paraphrase instructions. Instead, as we stated in our manuscript, *"we pre-defined a diverse prompt template pool to ensure that the LLMs are not overfit to a limited set of prompt formats"*. The template pool is shown in Table 5, 7, and 8 in our manuscript.
>
> Each instruction is generated by automatically filling these templates. This approach guarantees that the semantic relationship between the instruction and the target molecule remains precise and consistent by design.
>
> We will make this point clearer in the revised manuscript to prevent any confusion. Specifically, we rename the Table 5, 7, and 8 from "Prompt Templates for MolEdit/MolOpt/MolCustom" to "Instruction Template Pool for MolEdit/MolOpt/MolCustom".
>
> We are happy to further clarify if this concern remains.

---

> ### Author Response · Authors · 2025-11-21
>
> >W5 & Q2: and the benchmark focuses primarily on organic drug-like molecules, limiting generalization to other domains.
> >Distributional bias of molecular sources
> OpenMolIns is built mainly from ZINC, ChEMBL, and MOSES, which are biased toward drug-like molecules. Could this limit generalization to other chemical domains (e.g., materials or catalysts)?
>
> #### Our Response:
> Thank you for raising this important point regarding the molecular scope and molecular sources of S²-Bench. We agree that our current focus on organic, drug-like molecules from Zinc-250K can be a limitation that affects its generalization to other chemical domains like materials science. We value this chance to provide additional analyses and clarifications.
>
> **Firstly**, our choice of Zinc-250K is primarily motivated by its wide adoption in molecular generation research, well-curated molecular structures, and balanced coverage of drug-like chemical space, which together provide a standardized and computationally tractable benchmark for fair and reproducible evaluation.
>
> To assess whether Zinc-250K sufficiently represents real discovery-scale chemistry, we compared its distribution with a large reference database, PubChem with 9 million diverse molecules. Specifically, we analyzed the average (i) atom count, (ii) ring and branch count, and (iii) path lengths of the two datasets for comparison.
> | Dataset   | # Samples | Avg Atom Count | Avg Ring Count | Avg Branch Count | Avg Path Length |
> |-----------|-----------|----------------|----------------|------------------|-----------------|
> | Zinc-250K | 250,000   | 23.15          | 2.76           | 7.31             | 12.48           |
> | PubChem   | 9,000,000 | 25.18          | 2.8            | 7.87             | 13.07           |
>
> Across all these axes, Zinc-250K exhibits broadly similar statistical trends to PubChem, suggesting that Zinc-250K captures the core characteristics of drug-like chemistry and remains a reasonable proxy for discovery-scale small molecules, even if it does not encompass every extreme or domain-specific motif.
>
>
> **Second**, our decision to concentrate on drug-like molecules was driven by two primary factors:
> 1. Drug discovery remains one of the most prominent and high-impact applications for AI-driven molecular generation. Focusing on this area allows our benchmark to address the most immediate needs of the community.
> 2. The chemical space of drug-like molecules is supported by the most extensive, mature, and well-curated databases (e.g., ChEMBL, ZINC). This provided a solid and reliable foundation for constructing a high-quality benchmark.
>
> We see our current work as a critical and necessary first step. By establishing a rigorous evaluation framework in this well-studied domain, we create a solid foundation and methodology that can be extended to other areas. While we acknowledge that the direct generalizability to other domains is limited, we believe that a model’s ability to reason about drug-like molecules serves as a strong indicator of its underlying potential for general chemical understanding.
>
> Acknowledging this limitation, we are actively planning to expand our benchmark. Our future work will focus on incorporating diverse molecular datasets, such as inorganic molecules, polymers, and catalysts, to enhance the benchmark’s generalizability and utility for the broader chemistry community.
>
> We hope this expanded clarification addresses your concerns and demonstrates our commitment to the long-term development of this benchmark.

---

> ### Author Response · Authors · 2025-11-21
>
> >Q4. Language understanding vs. template learning
> Since instructions are generated from deterministic templates, does S²-Bench truly measure language understanding or just template matching? Have the authors tested models on non-templated, free-form instructions?
>
> #### Our Response:
> Thank you for raising this important point about whether S²-Bench evaluates genuine language understanding or merely template matching.
>
> First of all, we believe distinguishing true language understanding from template matching remains an open question for Large Language Models.
>
> In the design of our benchmark, we made deliberate efforts to mitigate template overfitting:
> 1. We explicitly adopt a multi-task setup. Different subtasks have distinct input formats and requirements, making it difficult for LLMs to rely on a single repetitive template. This examines the underlying task semantics rather than superficial patterns.
> 2. For each subtask, we pre-define a diverse template pool to reduce the likelihood that models memorize specific instructions, directly addressing concerns about "template learning".
>
> We also include an additional case study regarding non-templated, free-form instructions.
>
> > Case 1: Help me add a benzene ring to CCCOCCC.
> Generated Molecule: ICCCOCCCc1ccccc1
> Expert Remark: The model satisfied the request to append a benzene ring to CCCOCCC by adding an aromatic c1ccccc1 fragment tethered to the terminal alkyl chain. However, it also introduced an iodine atom (I) at the opposite terminus (ICCCO...), which was not requested.
>
> > Case 2: Help me remove the hydroxyl in CCCCCCO.
> Generated Molecule: CCCCCC
> Expert Remark: The model correctly removed the terminal hydroxyl from CCCCCCO, producing unfunctionalized hexane (CCCCCC). Structural change is minimal and valences are correct.
>
> > Case 3: Based on the molecule CCCOCCC, help me find a molecule with lower LogP value.
> Generated Molecule: O=C(O)COCCC
> Expert Remark: The model reduces the LogP value by introducing a carboxyl group, replacing part of the alkyl–ether motif with the more polar O=C(O)C– fragment. This substitution increases hydrogen-bonding capacity and overall polarity, which effectively lowers lipophilicity while keeping the structure chemically plausible and valence-consistent.
>
> Here, we test our OpenMolIns-tuned model with non-templated, free-form instructions. The results show that the model can reliably interpret open-ended queries and generate chemically valid molecules that satisfy the specified constraints, demonstrating strong generalization beyond the template learning.
>
> We truly appreciate your input, and we are committed to further expanding our instruction template pool to even further minimize the potential for instruction overfitting.

---

> ### Author Response · Authors · 2025-11-21
>
> >Q5. Out-of-distribution generalization
> Have the authors evaluated the benchmark under compositional or OOD instructions, e.g., “Add a hydroxyl group while keeping molecular weight below 200”?
>
> #### Our Response:
> Thank you for this insightful question about out-of-distribution (OOD) generalization. The example you provided, “Add a hydroxyl group while keeping molecular weight below 200,” is an excellent example of a compositional, multi-constraint instruction.
>
> We agree that evaluating a model’s ability to synthesize various constraints into a single, complex instruction is a crucial test of its true reasoning and generalization capabilities.
>
> To address this important point, we have conducted a new case study to probe our OpenMolIns-tuned models for the OOD and compositional generalization on a small set of challenging, multi-constraint instructions:
>
> >Case 1: The molecule contains 1 benzene rings group, and 2 aldehyde groups. Keep the logP value below 3.
> Generated Molecule: O=Cc1ccc(\[S-\])c(C=O)c1
> Expert Remark: The generated structure correctly incorporates one benzene ring and two aldehyde groups as specified. However, the model introduces an additional thiolate substituent (–S⁻) on the aromatic ring, which was not part of the constraint. This anionic group substantially increases molecular polarity and can drastically lower the logP value, suggesting that the model may be manipulating ionizable substituents to meet the logP requirement rather than adhering strictly to the requested functional-group composition.
>
> >Case 2: Modify the molecule CC\(C\)Nc1cc(-n2ccnc2)ncn1 by adding a halo and increase the qed value.
> Generated Molecule: CC\(C\)Nc1cc(-n2ccnc2)nc(Cl)n1
> Expert Remark: The model correctly introduced a chloro substituent at the benzene ring, satisfying the structural requirement. This halogenation is a subtle modification that increases molecular lipophilicity moderately without significantly increasing molecular weight or rotatable bonds, which can improve binding-like descriptors considered in QED.
>
> >Case 3: The molecule consists of 2 hydroxyl groups, and 1 benzene rings group and the ring count is below 3.
> Generated Molecule: CC\(C\)\(C\)c1ccc(O)c(O)c1
> Expert Remark: The generated molecule correctly contains one benzene ring and two hydroxyl groups attached to the aromatic ring, satisfying all functional-group requirements. The tert-butyl substituent (CC\(C\)\(C\)-) adds steric bulk but does not introduce additional rings, keeping the ring count below 3 as specified.
>
> We find that models trained on OpenMolIns demonstrate strong potential to generalize to multi-constraint instructions. Remarkably, even for unseen properties, such as the number of rings, these models can comprehend and follow the intended semantics.
>
> #### References
> [1] Polykovskiy, D., Zhebrak, A., Sanchez-Lengeling, B., Golovanov, S., Tatanov, O., Belyaev, S., ... & Zhavoronkov, A. (2020). Molecular sets (MOSES): a benchmarking platform for molecular generation models. Frontiers in pharmacology, 11, 565644.

---

### Official Review · Reviewer_Eqbw · 2025-11-01

**Soundness:** 3
**Presentation:** 2
**Contribution:** 3
**Rating:** 6
**Confidence:** 3

**Summary:**

The paper introduces Speak-to-Structure, a benchmark for open-domain, one-to-many natural language–driven molecular design, comprising three task families, MolEdit, MolOpt, and MolCustom, which focus on precise editing, property-oriented optimization, and de novo constrained generation. It also releases OpenMolIns, an instruction-tuning dataset and reports results across 28 LLMs, showing that targeted one-to-one datasets overestimate real design ability while instruction-tuned models on OpenMolIns perform best.

**Strengths:**

1. The benchmark allows many valid molecules per prompt, not just one “right” answer, which is closer to how chemists actually design.
2. Success is checked automatically, and everything rolls up into a single headline number (WSR) so models are straightforward to compare.
3. Many models are evaluated side-by-side, revealing where current methods struggle (especially de-novo constraints) and showing that instruction-tuning on the released data can meaningfully boost performance.

**Weaknesses:**

1. The “weighted success rate” is computed as *success × one quality term* (similarity for MolEdit/MolOpt; novelty for MolCustom), then averaged uniformly across nine subtasks. Because there are no reported thresholds or sensitivity analyses, rankings may be unstable under this multiplicative choice and the equal subtask weights.
2. Prompts are generated from fixed templates, and MolCustom’s constraints largely boil down to counts of atoms, bonds, or functional groups. Important real-world specs—stereochemistry, 3D geometry, ring/bridge topology, and basic synthesizability—are not directly assessed.
3. The test molecules are drawn from Zinc-250K for convenience, and the builders pre-select which functional groups/atoms/bonds occur, including “normal/edge” subsets. These design choices can shift task difficulty and may not reflect the breadth of discovery-scale chemical space.

**Questions:**

Please refer to the weaknesses.

---

> ### Author Response · Authors · 2025-11-17
>
> Thank you so much for your valuable comments! We are glad to hear that you acknowledge the importance of text-based open-domain molecular generation, our automatic evaluation process, and our comprehensive comparison of various LLMs. We cherish the chance to resolve your concerns in details:
>
> >W1: The “weighted success rate” is computed as success × one quality term (similarity for MolEdit/MolOpt; novelty for MolCustom), then averaged uniformly across nine subtasks. Because there are no reported thresholds or sensitivity analyses, rankings may be unstable under this multiplicative choice and the equal subtask weights.
>
> #### Our Response:
> Thank you for your valuable comments. We acknowledge your concern regarding the quality terms. Here, we take several steps to verify whether WSR aligns with human expert judgements.
>
> **First of all**, we aim to clarify that the quality term ensures chemically meaningful success rather than merely passing the automatic testing process.
>
> In the **MolCustom** task, novelty is a standard evaluation metric that has been widely adopted in prior molecular generation studies. The multiplication of novelty x success\_rate is meaningful, as it ensures that the reported performance reflects both the correctness of the generation and its ability to produce genuinely new molecules.
>
> In **MolEdit** and **MolOpt** tasks, the models are allowed to make extra modifications to ensure the chemical validity, while the exact edit path on SMILES strings cannot be traced. Therefore, the similarity term helps verify whether the model’s generation is truly derived from the input molecule or an unrelated structure.
>
> Our correlation analysis focused on the performance of BioT5-base in MolEdit and MolOpt, where the model achieves high success rates but performs poorly on the quality term, indicating that the generated molecules are not modified from the original molecule but from scratch. For this analysis, we first randomly sampled 100 examples from each subtask and then asked human experts to evaluate the correctness and quality of the generated molecules. Each example is assigned an expert score from 0 to 5, where:
> * 5 denotes a perfect success,
> * 4 means success but with redundant operations (a little different structure),
> * 3 shows failure but modified from the original molecule (similar structure),
> * 2 denotes failure and a little different structure,
> * 1 indicates success but unrelated/different structure,
> * 0 denotes failure and unrelated/different structure.
>
> We calculate the average score of expert evaluation. The results are shown in the table below:
>
> | BioT5-base |    Subtasks          | SR     | Similarity | WSR    | Validity | Expert Evaluation |
> |------------|--------------|--------|------------|--------|----------|----------|
> | MolEdit    | AddComponent | 0.3462 | 0.1567     | 0.0542 | 1        | 0.64        |
> |            | DelComponent | 0.1668 | 0.1597     | 0.0266 | 1        | 0.22        |
> |            | SubComponent | 0.0684 | 0.1576     | 0.0108 | 0.9998   | 0.19        |
> | MolOpt     | LogP         | 0.5158 | 0.1526     | 0.0787 | 1        | 0.58        |
> |            | MR           | 0.506  | 0.1597     | 0.0808 | 1        | 0.67        |
> |            | QED          | 0.5068 | 0.158      | 0.0801 | 1        | 0.64        |
>
> We computed both Pearson and Spearman correlations between SR/WSR and the expert evaluation scores across the subtasks. The results show that WSR exhibits consistently stronger alignment with expert ratings. Specifically, WSR achieves a slightly higher Pearson correlation with expert scores (r = 0.926 vs. 0.922 for SR) and, more importantly, a substantially higher Spearman rank correlation (ρ = 0.899 vs. 0.551 for SR). While the correlation between SR and expert ratings is not statistically significant (p = 0.257), WSR shows strong and significant rank consistency (p = 0.015).
>
> Therefore, it is essential to have these quality terms to reflect the genuine capabilities of LLMs more faithfully.
>
> **Finally**, regarding equal subtask weights, as we treat all the major tasks and subtasks equally important, we believe it reasonable to adopt equal weights to form a balanced benchmark.
>
> We sincerely hope that our clarification towards the quality terms and subtask weights could resolve your concerns.

---

> ### Author Response · Authors · 2025-11-17
>
> >W2: Prompts are generated from fixed templates, and MolCustom’s constraints largely boil down to counts of atoms, bonds, or functional groups. Important real-world specs—stereochemistry, 3D geometry, ring/bridge topology, and basic synthesizability—are not directly assessed.
>
> #### Our Response：
> Thank you for your valuable suggestion! We fully agree that stereochemistry, 3D geometry, ring/bridge topology, and basic synthesizability represent important aspects of real-world molecular design.
>
> In our design, we include the counts of atoms, bonds, and functional groups to prob LLMs' understanding towards the molecular syntax, especially the details of the molecular structure.
>
> However, we agree that incorporating such complex real-world constraints would indeed make the benchmark more realistic. Specifically, we could consider the following tasks:
>
> 1. Open stereochemistry–aware generation: Generate any valid molecule that satisfies a set of high-level stereochemical requirements (e.g., “one chiral center, R configuration; one E double bond; no fixed scaffold”).
> 2. Open 3D geometry–constrained generation: Generate molecules whose 3D pharmacophore distances and angles meet coarse geometric constraints (e.g., “two hydrogen-bond donors 8–10 Å apart”).
> 3. Open topology- or scaffold-pattern generation: Generate any structure that matches only a topological pattern, such as “one spiro junction + one 6–5 fused system,” without specifying the exact ring identities or substituent arrangement.
> 4. Open synthesizability–aware generation: Generate molecules that meet only coarse synthetic constraints (e.g., "SA score < 4" or "≤ 2 disallowed motifs").
>
> In the revised manuscript, we have discussed this as a current limitation. We appreciate your suggestion and view this as an exciting and important future direction. We are committed to extend our benchmark with additional real-world requirements.
>
> >W3: The test molecules are drawn from Zinc-250K for convenience, and the builders pre-select which functional groups/atoms/bonds occur, including “normal/edge” subsets. These design choices can shift task difficulty and may not reflect the breadth of discovery-scale chemical space.
>
> #### Our Response:
> Thank you for this insightful comment. We agree that relying on Zinc-250K and pre-selecting functional groups/atoms/bonds could potentially constrain chemical diversity and shift task difficulty.
>
> Our choice of Zinc-250K is primarily motivated by its wide adoption in molecular generation research, well-curated molecular structures, and balanced coverage of drug-like chemical space, which together provide a standardized and computationally tractable benchmark for fair and reproducible evaluation.
>
> Nevertheless, we agree that expanding the chemical space would further enhance realism. As part of our ongoing work, we plan to:
> (i) enlarge the rule base to include less frequent functional groups, more complex heterocycles, and diverse ring topologies;
> (ii) incorporate non-traditional molecular domains, including inorganic and materials-oriented compounds, whose structural patterns differ substantially from drug-like molecules.
>
> We report molecular distribution statistics for the original Zinc-250K, PubChem, and our pre-selected samples used in MolEdit and MolOpt. Specifically, we compare the average (i) atom count, (ii) ring count, (iii) branch count, and (iv) path length across datasets.
> | Dataset   | # Samples | Avg Atom Count | Avg Ring Count | Avg Branch Count | Avg Path Length |
> |-----------|-----------|----------------|----------------|------------------|-----------------|
> | Zinc-250K | 250,000   | 23.15          | 2.76           | 7.31             | 12.48           |
> | PubChem   | 9,000,000 | 25.18          | 2.8            | 7.87             | 13.07           |
> | S²-Bench's sampling   | 30,000 | 23.17          | 2.74            | 7.34             | 12.47           |
>
> Across all these axes, Zinc-250K exhibits broadly similar statistical trends to PubChem, suggesting that Zinc-250K captures the core characteristics of drug-like chemistry and remains a reasonable proxy for discovery-scale small molecules. Furthermore, the distribution of our pre-selected samples aligns closely with the original Zinc-250K, suggesting that our pre-selection does not substantially distort the underlying molecular distribution.
>
> Finally, we sincerely appreciate your insightful comments. We believe these extensions will surely better align our benchmark with real-world molecular discovery pipelines.

---

### Author Response · Authors · 2025-11-26
**Waiting for further discussion**

Dear Reviewer,

Thank you again for your detailed and thoughtful review. We wanted to let you know that we have now provided a detailed response with additional experiments to address the points you raised, which has helped us significantly improve the paper and provide stronger evidence for our central claims.  We also welcome any further feedback you might be willing to share. Your insights are invaluable to us, and we're eager to address any remaining issues to improve our work.

We sincerely appreciate your time and your constructive feedback!

---

### Meta-Review · Area_Chair_jq2D · 2026-01-06

**Summary:**

This paper releases a benchmark meant to test open-domain with one instruction to many valid molecules inmolecular design. The benchmark is splited into MolEdit (localized edits), MolOpt (property-oriented edits), and MolCustom (de novo constrained generation). It also releases a large instruction-style dataset (OpenMolIns) and evaluates 28 LLMs, arguing that classic one-to-one text-to-SMILES datasets can overestimate true design ability, and that instruction-tuning on OpenMolIns yields the strongest performance.

The reviewers appreciate the workflow and the uniqueness of the one-to-many design and the board model sweep. However, the reviewers are still concerned about the methodological depth and the novelty of this paper. The reviewers are also concerned about the task realism as well as the representativeness of the selected dataset, which makes this paper does not meet the bar of acceptance.

**Reviewer Concerns:**

The reviewers are satisfied with their concern regarding the WSR, and the additionally provided experiments. However, several works are still required to fully address the reviewer's concern.
1. Regarding the WSR robustness (Reviewer Eqbw). Although the authors have provided some human evaluations, more quantitive results on the robustness is expected.
2. Missing real-world constraints and the data representation of ZINC-250K dataset from Reviewer Eqbw remains open. As a benchmark work, the authors are expected to provide more data and realistic constraints. Similar question is also raised by Reviewer fcDQ.
3. Non-templated evaluation, as requested by Reviewer fcDQ, is important to evaluate the LLM's ability for free-form generation. This is still missing in the rebuttal.

**Reviewer Scores:**

The first two reviewers Eqbw and fcDQ already give a good score and is unlikely to be higher with some open question unaddressed. Reviewer Cos8 had some discussion with the authors on the novelty / related works and it seems that the authors response have not fully addressed their concern. Nevertheless, given the remaining open questions, the reviewer's update will not significantly change the status of the decision.

---

### Decision · Program_Chairs · 2026-01-26

Reject